# A kinetic model predicts $Sp$Cas9 activity, improves off-target classification, and reveals the physical basis of targeting fidelity

Behrouz Eslami-Mossallam[1,6,10], Misha Klein[1,7,10], Constantijn V. D. Smagt [1,7], Koen V. D. Sanden [1], Stephen K. Jones Jr. [2,3,4,8], John A. Hawkins [2,3,4,5,9], Ilya J. Finkelstein [2,3,4] & Martin Depken [1✉]

The *S. pyogenes* (*Sp*) Cas9 endonuclease is an important gene-editing tool. *Sp*Cas9 is directed to target sites based on complementarity to a complexed single-guide RNA (sgRNA). However, *Sp*Cas9-sgRNA also binds and cleaves genomic off-targets with only partial complementarity. To date, we lack the ability to predict cleavage and binding activity quantitatively, and rely on binary classification schemes to identify strong off-targets. We report a quantitative kinetic model that captures the *Sp*Cas9-mediated strand-replacement reaction in free-energy terms. The model predicts binding and cleavage activity as a function of time, target, and experimental conditions. Trained and validated on high-throughput bulk-biochemical data, our model predicts the intermediate R-loop state recently observed in single-molecule experiments, as well as the associated conversion rates. Finally, we show that our quantitative activity predictor can be reduced to a binary off-target classifier that outperforms the established state-of-the-art. Our approach is extensible, and can characterize any CRISPR-Cas nuclease – benchmarking natural and future high-fidelity variants against *Sp*Cas9; elucidating determinants of CRISPR fidelity; and revealing pathways to increased specificity and efficiency in engineered systems.

[1] Kavli Institute of NanoScience and Department of BionanoScience, Delft University of Technology, Delft 2629HZ, the Netherlands. [2] Department of Molecular Biosciences, University of Texas at Austin, Austin, TX 78712, USA. [3] Institute for Cellular and Molecular Biology, University of Texas at Austin, Austin, TX 78712, USA. [4] Center for Systems and Synthetic Biology, University of Texas at Austin, Austin, TX 78712, USA. [5] Oden Institute for Computational Engineering and Science, University of Texas at Austin, Austin, TX 78712, USA. [6] Present address: Dept. Building Physics and Systems, TNO Building and Construction Research, Leeghwaterstraat 44, Delft, The Netherlands. [7] Present address: Department of Physics and Astronomy, and LaserLaB Amsterdam, Vrije Universiteit Amsterdam, De Boelelaan 1081, 1081 HV Amsterdam, the Netherlands. [8] Present address: VU LSC-EMBL Partnership for Genome Editing Technologies, Life Sciences Center, Vilnius University, Vilnius, Lithuania. [9] Present address: European Molecular Biology Laboratory, Genome Biology Department, Heidelberg, Germany. [10] These authors contributed equally: Behrouz Eslami-Mossallam, Misha Klein. ✉email: S.M.Depken@tudelft.nl

CRISPR-Cas9 (Clustered Regularly Interspaced Short Palindromic Repeats—CRISPR-associated protein 9) has become a ubiquitous tool in the biological sciences[1,2], with applications ranging from live-cell imaging[3] and gene knock-down/overexpression[4,5] to genetic engineering[6,7] and gene therapy[8,9]. *Streptococcus pyogenes* (*Sp*) Cas9 can be programmed with a ~100 nucleotide (nt) single-guide RNA (sgRNA) to target DNAs based on the level of complementarity to a 20 nt segment of the sgRNA[10]. Wildtype *Sp*Cas9 (henceforth Cas9) induces site-specific double-stranded breaks and the catalytically dead Cas9 (dCas9) mutant allows for binding without cleavage[3,5]. Apart from complimentary on-targets, Cas9-sgRNA also binds and cleaves non-complementary off-targets[11–18]. Off-target cleavage risks deleterious genomic alterations, which has so far impeded the widespread implementation of the CRISPR toolkit in human therapeutics[19].

Strong off-target sites are identified in silico by a growing set of tools. These tools use bioinformatics[20,21], machine learning[22,23], or heuristic[12,14,24,25] approaches to rank genomic sites based on distinctive off-target activity scores. Though such models can identify strong off-targets, they are not quantitative and cannot assess activity on the many lesser off-targets; nor can they predict how activity changes with exposure time and enzyme concentration—even though these parameters are frequently exploited to limit off-target activity in cells[26].

To implement quantitative activity prediction, Cas9 targeting must be modelled in physical terms. Existing physical models[24,27,28] assume binding equilibration before cleavage, and it remains unclear what predictive power such approaches can ultimately deliver in this non-equilibrium system[29,30]. To account for the nonequilibrium nature of the targeting reaction, we construct a mechanistic model that captures binding and cleavage reactions in kinetic terms. To gain insights into general mechanisms, we train and validate our model on high-throughput datasets that capture both binding and cleavage in bulk experiments[15,31]. Though we restrict our training to off-targets with two or less mismatches, we accurately predict the activities on all more highly mismatched off-targets in the same datasets, as well as those reported in two independent high-throughput datasets[11].

To reveal the physical basis of Cas9 fidelity on genomic scales, we extract the free-energy landscapes that control PAM binding, strand-replacement, and cleavage on any target. Our characterization of Cas9 supports the notion that observed differences in binding and cleavage activities[32–41] stem from a relatively long-lived DNA-bound RNA-DNA hybrid (R-loop) intermediate. This R-loop intermediate was recently observed directly in single-molecule experiments[42], and our model predicts both its location and its conversion rates.

Though the strengths of our model lies in that it allows us to calculate how (d)Cas9 activity evolves in time under various conditions, we also sought to compare our approach to existing binary off-target classifiers that identify strong off-targets. To this end, we reduce our quantitative activity predictor to a binary off-target classifier that outperforms the leading tools used today[12,24,28,43].

## Results

**The kinetic model**. In Fig. 1a we show the reaction pathway that underpins the Cas9 targeting reaction on every target[44]. The reaction starts with Cas9-sgRNA ribonucleoprotein complex exiting the solution state to specifically bind to a 3nt protospacer adjacent motif (PAM) DNA sequence—canonically 5'-NGG-3'—via protein-DNA interactions[44,45]. Binding to the PAM sequence (state 0) opens the DNA double helix, and allows the first base of

the target sequence to hybridize with the sgRNA[44,45], forming the first R-loop state (state 1). The DNA double helix further denatures as the RNA-DNA hybrid is extended in the guide-target strand-replacement reaction[46–49] (state 2-20). The hybrid grows and shrinks in single-nucleotide steps, until it is either reversed and Cas9 dissociates, or it reaches completion at 20 base pairs (bp) in state 20. If the full hybrid is formed, Cas9 can use its HNH and RuvC nuclease domains to cleave both DNA strands[50].

If we know the conversion rates in Fig. 1a for a particular guide and target, the reaction scheme can be solved to calculate the binding and cleavage probabilities at any time (Methods). Fully parameterizing the model over all guide and target sequences requires the estimation of ~$10^{26}$ rates. To render parameter estimation tractable, we make four mechanistic-model assumptions:

(1) Mismatch positions are more important than mismatch types (e.g. G-G vs. G-A). This can be directly inferred from data[11,15], and we treat all 12 mismatch types equally.

(2) Mismatch energies are determined by local interactions. The energetic cost of multiple mismatches is taken to be equal to the sum of the energetic costs of the individual mismatches.

(3) dCas9 differs from Cas9 only in that dsDNA bond-cleavage catalysis is completely suppressed ($k_{cat} = 0$); all other rates are taken to be identical[51,52].

(4) All selectivity is governed by the hybrid-bond-reversal rates. Hybrid-bond-formation rates are treated as equal, independent of complementarity and location.

These assumptions reduce the total number of microscopic parameters to 44 (see Methods): the (concentration dependent) rate of PAM binding from solution ($k_{on}$) and the associated free-energy gain ($F_0$); a single internal forward bond-formation rate ($k_f$); 20 free energies dictating R-loop progression at the on-target ($F_1, \ldots, F_{20}$); 20 free-energy penalties for mismatches at different R-loop positions ($\delta\epsilon_1, \ldots, \delta\epsilon_{20}$); and the rate at which the final cleavage reaction is catalyzed for Cas9 ($k_{cat}$). Once model parameters are estimated, all possible off-target free energies can be directly calculated using assumptions 1–4 above. In Fig. 1b we illustrate the calculation taking us from the on-target (pink) to the off-target (blue) free-energy landscape with mismatches entering the hybrid at the 3rd and 15th bp. How to translate between free energies and rates is detailed in Methods.

Base-pairing interactions, protein-DNA interactions[52], and induced conformational changes[50,51,53,54] all contribute to the stability of the Cas9-sgRNA-DNA complex. To account for the varying nature of these interactions, we allow for varying gains and losses in the on-target free-energy landscape as the hybrid is extended. These variable gains and losses allow for the formation of metastable states on the on-target, and constitutes an essential extension of our previous fixed-gain model for RNA-guided nuclease kinetics[30], as well as of models describing DNA displacement reactions occurring in solution[55–58].

**Training on binding and cleavage for moderately mismatched targets**. We seek to reveal general properties of *Sp*Cas9 DNA targeting on genomic scales. To this end, we train and validate our model on data from two highly reproducible bulk-biochemical experiments performed on a large library of moderately to highly mismatched off-targets. The first set[15] (NucleaSeq) has 97% correlation between replicated experiments, and estimates the effective cleavage rates ($k_{clv}^{eff}$) for a library of off-targets exposed to Cas9-sgRNA for 16 hours. The second set[15,31] (CHAMP) has 94% correlation between replicated experiments, and reports on the effective association constant ($K_A^{eff}$) over the same library and guide, but this time exposed to dCas9-sgRNA

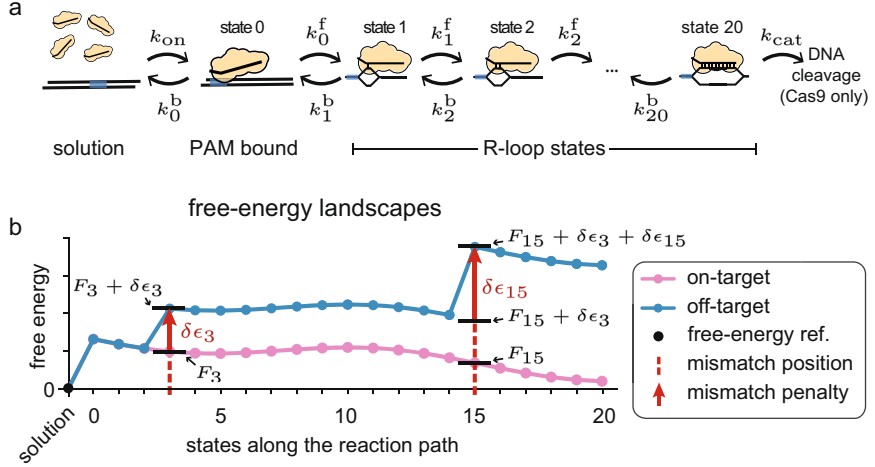

**Fig. 1 The reaction scheme and the implications of the model assumptions. a** The general microscopic reaction scheme for PAM (blue rectangle) binding from solution, followed by strand replacement and eventual cleavage (Cas9 only). The bound states are labeled 0-20, starting with the PAM bound state, and ending with the state having a fully open R-loop (20 bp hybrid). **b** An example on-target free-energy landscape $F_n$ (pink), and the resulting free-energy landscape when using our mechanistic-model assumptions on an off-target where mismatches enter the hybrid at length 3 and 15 bp (blue). Each mismatch (dashed red line) has an energetic cost $\epsilon_n$ (red arrow) added onto the free energy of all later R-loop states. The solution state is chosen as a reference for the free energy, and set to $0k_BT$ (black point).

for 10 min. In Methods we detail how the experiments are modeled.

We estimate the model parameters by minimizing the total experimental-error weighted residue between prediction and experiment for off-targets (see Methods) with no more than two mismatches in the NucleaSeq (Fig. 2a–c) and CHAMP (Fig. 2d–f) experiments. The rates and association constants from different types of mismatches are averaged (see Methods and Supplementary Data 1), and the optimal solution is sought with a Simulated Annealing algorithm[59] (see Methods).

The two training sets differ significantly (Fig. 2, and Supplementary Fig. 1a). Our model still reproduces effective cleavage rates (Fig. 2a–c) and effective association constants (Fig. 2d–f) with a Pearson correlation of 93% and 98% respectively, and quantitatively captures the difference between binding and cleavage activity. The time and concentration dependence of (d)Cas9 activity can be explored through a dashboard we provide (see Code Availability).

**Validation on highly mismatched targets and independent data sets.** Apart from the data we use for training (two or less mismatches), the NucleaSeq[15] and CHAMP[15,31] sequence libraries also includes block-mismatched targets with more than two mismatches. In Fig. 3a, b we show that we quantitatively predict effective association constants on these highly mismatched targets at a correlation of 98%. Our method also successfully separates out the single dominating off-target present among highly mismatched targets in the NucleaSeq experiments (Supplementary Fig. 1b), resulting in a perfect correlation.

To further validate our model, we test against two data sets from HiTS-FLIP experiments reported in the literature[11]. The first independent validation set records the association rate relative to the on-target, estimated over 1500 seconds of exposure to dCas9-sgRNA at 1 nM concentration (Fig. 3c–e). The second independent validation set records the dissociation rate relative to the on-target, estimated over 1500 seconds following 12 hours of exposure to a saturating dCas9-sgRNA concentration (Fig. 3f–h). Our model quantitatively captures the relative association rates for all reported targets with 82% correlation (Fig. 3e). For the relative dissociation rates, the correlation is more modest at 46% (Fig. 3h), and the quantitative agreement is lost in some regions (Fig. 3f–h). We still

seem to capture the general trends on moderately mismatched targets (Fig. 3f, g), though our model will never give binding/dissociation rates above/below that of the on-target, as is reported for some highly mismatched targets (Fig. 3e, h)

**Physical characterization of SpCas9 and the intermediate R-loop state.** As our model parameters carry physical meaning, estimating them from data amounts to characterizing the system in physical terms. For Cas9, it has been experimentally shown that R-loop progression is controlled by an intermediate meta-stable state on the on-target[42]. We expect this intermediate state to show up as a local minimum in our estimated on-target free-energy landscape. The free energy of any metastable state will have a strong influence on the observed dynamics, and we expect such energies to be well constrained by the data. We expect barriers between metastable states to be harder to resolve, as the details of barrier regions matter less for the observable dynamics.

We here report 33 near-equivalent optimization runs that all resulted in a residue that fell within 15% of the best solution found (see Supplementary Video 1). In Fig. 4a we plot the resulting on-target free-energy landscapes, with the optimal solution highlighted in pink. As expected, we see metastable states in the on-target free-energy landscape. With Cas9 in solution or PAM-bound, we have a well-defined free-energy minimum where the R-loop is closed (C). The on-target free energy (Fig. 4a) increases substantially when forming the first hybrid bp in state 1, and remains relatively high and poorly constrained up to and including state 8. The energy of state 9-12 are well constrained, and among them we find a second local minimum. We identify these states as belonging to an intermediate (I) R-loop state. For hybrids of length 13 to 19 bp we again see an ill-constrained barrier, ending when we enter a well-constrained local minimum of a fully formed hybrid at state 20. This last minima defines the open (O) R-loop.

Mismatch penalties are all around $5k_B T$ (Fig. 4b), but show reproducible variation along the hybrid. Comparing Fig. 2a, d with Fig. 4b, it is clear that variations in mismatch penalties in the first 8 states correlate strongly with the measured effective cleavage rate/dissociation constant on targets with a single seed mismatch at the corresponding hybrid position. It is not clear if these variations are due to varying interactions with the protein,

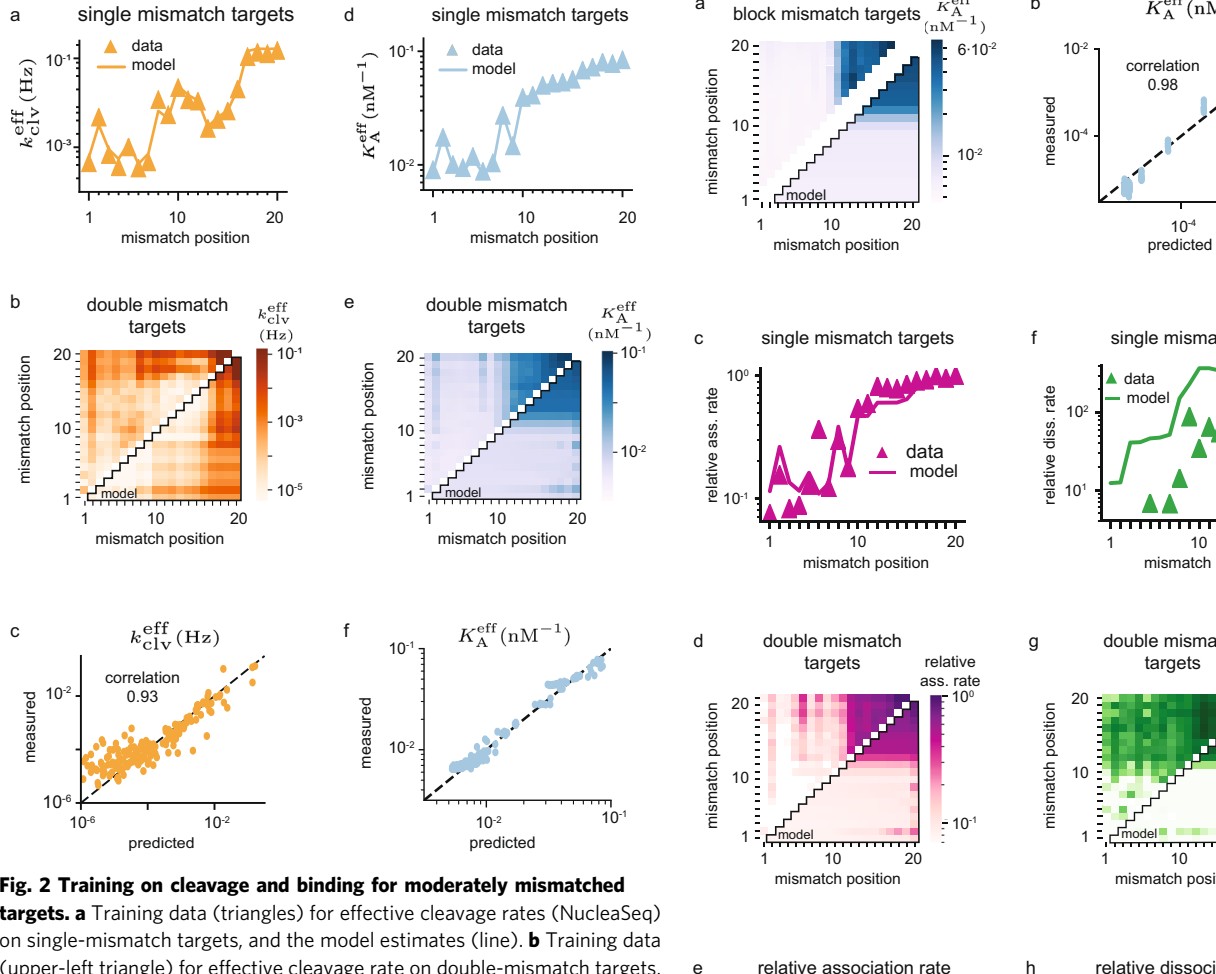

**Fig. 2 Training on cleavage and binding for moderately mismatched targets. a** Training data (triangles) for effective cleavage rates (NucleaSeq) on single-mismatch targets, and the model estimates (line). **b** Training data (upper-left triangle) for effective cleavage rate on double-mismatch targets, and the model estimates (lower-right triangle). **c** Correlation plot for all effective cleavage rate data used for training (single- and double-mismatch targets). **d** Training data (triangles) for effective association constant (CHAMP) on single-mismatch targets, and the model estimates (line). **e** Training data (upper-left triangle) for effective association constant on double-mismatch targets, and the model estimates (lower-right triangle). **f** Correlation plot for all effective association constant data used for training (single- and double-mismatch targets). All data is averaged over mismatch type (see Supplementary Data 1). The quoted correlation coefficients are Pearson-correlation coefficients, and correlation plots are displayed with log-scales to show the quantitative agreement also for weak targets. The dashed line in the correlation plots correspond to perfect quantitative prediction.

or reflects the fact that the possible mismatch types vary with position. In Fig. 4c we show the remaining rates needed to predict Cas9 cleavage activity at any target, time, and Cas9-sgRNA concentration (see Methods).

**R-loop dynamics captures single-molecule experiments.** The recent direct observation of the R-loop dynamics between meta-stable states[42] allows us to further test our model against quantitative single-molecule data. To this end, we define a coarse-grained model (Fig. 5a) and calculate the effective rates between metastable states from our microscopic free-energy landscapes (see Methods). In Supplementary Fig. 2 we show that predictions based on our coarse-grained model replicate those of the microscopic model.

Using effective rates between metastable states, we can rationalize the broad strokes of Cas9 fidelity by considering a few important examples[42]. For on-targets (Fig. 5b), the transition between the

PAM bound state and the intermediate R-loop state is reversible ($k_{PI} \approx k_{IP}$) (Fig. 5c). Complexes that enter the intermediate state typically also enter the fully opened state ($k_{IP} \ll k_{IO}$). The transition from intermediate to open R-loop configuration is irreversible ($k_{IO} \gg k_{OI}$), and entering the open configuration guarantees cleavage ($k_{OI} \ll k_{cat}$). Taken together, the on-target reaction is essentially unidirectional toward cleavage, once the intermediate state is entered. The transition into the intermediate R-loop state is rate-limiting ($k_{PI} \ll k_{IO} \ll k_{cat}$) for cleavage.

Mismatches between the target DNA and the sgRNA have differential effects on R-loop propagation depending on position. A PAM-proximal mismatch (position 1–8) (Fig. 5d) strongly suppresses the rate of transition from a closed to intermediate R-loop state (Fig. 5e). In contrast, a PAM-distal mismatch (position 12–17) (Fig. 5f) limits the effective rate of cleavage by reducing the intermediate to open transition rate (Fig. 5g), and allowing for re-closure of the R-loop before entering the open state ($k_{IO} \approx k_{IP}$).

These observations are in agreement with the experimental observation[42], and in Fig. 5c, e we use purple triangles to indicate measured rates[42] when available at zero torque. We quantitatively

**Fig. 3 Validation on highly mismatched targets and independent HiTS-FLIP data. a** Validation data (upper-left triangle) for effective association constant (CHAMP) on block-mismatched targets, and model estimates (lower-right triangle). The two terminal mismatch positions in the block are marked on the axes. **b** Correlation plot between measured effective association constants and model predictions on block-mismatched targets. **c** Validation data (triangles) for association rates (HiTS-FLIP data set[11]) on single-mismatch targets, and model estimates (line). **d** Validation data (upper-left triangle) for association rates on double-mismatch targets, and model estimates (lower-right triangle). **e** Correlation plot for all positive association rates, including moderately (1–2 mismatches, dark purple) and highly (3–20 mismatches, light purple) mismatched targets. **f** Validation data (triangles) for dissociation rates (HiTS-FLIP data set[11]) on single-mismatch targets, and model estimates (line). The missing mismatch-averaged dissociation rates in the seed are negative. **g** Validation data (upper-left triangle) for dissociation rates on double-mismatch targets, and model estimates (lower-right triangle). **h** Correlation plot for all positive dissociation rates, including moderately (1–2 mismatches, dark green) and highly (3–20 mismatches, light green) mismatched targets. Mismatch-averaged rates dominated by negative scores are excluded from the analysis, and all data is averaged over mismatch type (see Methods and Supplementary Data 1). The quoted correlation coefficients are Pearson-correlation coefficients, and correlation plots are displayed with log-scales to show the quantitative agreement also for weak targets. The dashed lines in the correlation plots correspond to perfect quantitative prediction.

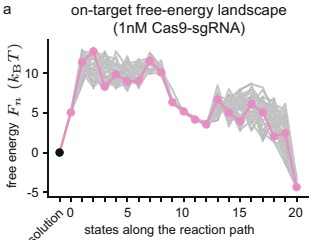
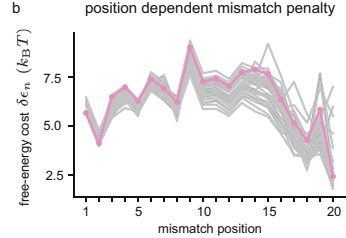
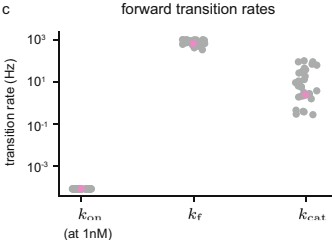

**Fig. 4 Physical parameters estimated from NucleaSeq and CHAMP datasets. a** The on-target free-energy landscape $F_n$ for (d)Cas9-sgRNA at the reference concentration 1 nM. The solution state (black dot) is taken as a reference for the free energy, and set to $0 k_B T$. State 0 is the PAM-bound state, and the remaining states are the R-loop states with hybrid length 1–20 bp. Three well defined local minima separated by barriers are visible, indicating that there are three meta-stable states in the system. **b** Energetic penalties $\delta\epsilon_n$ incurred by mismatches as a function of position $n$ in the hybrid. **c)** The estimates for the on-rate at 1 nM Cas9-sgRNA concentration ($k_{on}$), the internal forward rate ($k_f$), and the bond-cleavage catalysis rate ($k_{cat}$). In all figures, the 33 near-equivalent solutions (see text) are plotted in grey, with the optimal solution highlighted in pink (Supplementary Data 1).

predict the conversion rates out of the intermediate R-loop state. The model also captures the position of the on-target intermediate state as being around hybrid length 9-12. Our model does not capture the rate of the open to intermediate transition, and future work will have to determine if this is due to a difference in experimental conditions or because our choice of training data is ill-suited to determine the free energies past the intermediate state.

Our model predicts rates on all off-targets, and so extends and refines the long-established rule of thumb that off-target rejection in the PAM proximal seed requires only one mismatch, while off-target rejection outside the seed region requires multiple mismatches[10]. In particular, our model quantifies the intermediate activity resulting from PAM distal mismatch, and so enables prediction of activity titration.

**R-loop dynamics resembles conformational dynamics.** Next, we wondered what structural properties of Cas9 give rise to the free-energy landscape of Fig. 4a. A comparison between DNA-bound and unbound Cas9-sgRNA structures have revealed that Cas9 repositions its HNH and RuvC nuclease domains to catalyze cleavage[45,60,61]. Ensemble FRET experiments detected two dominant Cas9 conformers with distinct HNH states[50], and single-molecule FRET studies have identified a third intermediate conformer[51,53,54].

The relative position and occupancy of the HNH states is affected by R-loop mismatches[51,53,54], and Ivanov et al.[42] suggest that the intermediate R-loop state is linked to the intermediate structural state seen in FRET experiments[51]. To test this hypothesis, we mimicked the experiments of Dagdas et al.[51], and considered the time evolution of the occupancy of our metastable R-loop states for two target sequences (Fig. 6). The

HNH-domain completes its conformational change within seconds after Cas9-sgRNA binds to on-target DNA[51], and our model demonstrates a similar behavior for R-loop progression (Fig. 6a). The intermediate structural state is visited only transiently[51], as is the intermediate R-loop state in our model (Fig. 6a). Compared to the on-target, PAM-distal mismatches maintain the entry rate into the intermediate structural state, while increasing the time spent in this state[51]; again in close agreement with our findings for the intermediate and open metastable R-loop states in the presence of a PAM distal mismatch (Fig. 6b). Taken together, our model supports the notion that the intermediate R-loop state is linked to the intermediate structural state seen in FRET experiments.

**Kinetic modelling improves genome-wide off-target prediction.** Current methods[12,14,20–25,28,43] for identifying strong off-targets rank genomic sequences according to various measures of activity. They do not quantitatively predict biochemically measurable parameters, nor do they normally capture changes in conditions or activity over time. Our approach overcomes these limitations, and we do not suggest that these benefits should be abandoned in order to construct a binary off-target classifier. Still, to strengthen the case for including the full non-equilibrium nature of the problem in any Cas9 modelling, we reduce our quantitative kinetic model to a binary classifier (referred to as kinetic classifier) and test how well it performs against three established state-of-the-art off-target predictors: a recent benchmarking of models[28] shows the CRISPRoff classifier to outperform the competition, so we first test against this tool; second, we test against the more recent uCRISPR[24] tool, which is based on hybrid energetics and has not been tested against CRISPRoff; lastly, we test against the Cutting Frequency Determination

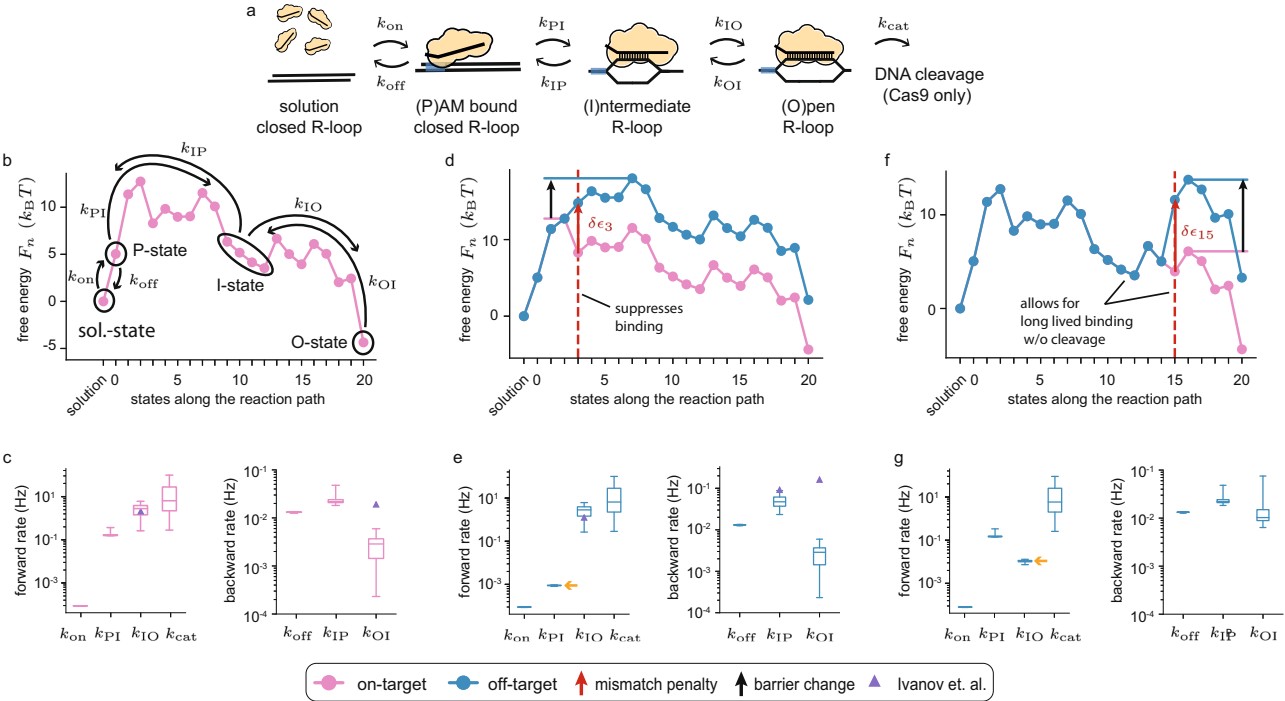

**Fig. 5 Metastable states control the targeting dynamics. a)** A coarse-grained version of the reaction scheme shown in Fig. 1a. Apart from the unbound and post-cleavage state, the targeting-reaction pathway is reduced to just three states: PAM bound and R-loop closed (0 bp hybrid), intermediate R-loop (7–13 np hybrid), and open R-loop (20 bp hybrid). **b** Microscopic free-energy landscape for the on-target exposed to 1 nM (d)Cas9-sgRNA (Fig. 4a) with coarse-grained states and rates indicated in black. **c** The calculated (see Methods) coarse-grained forward and backward rates on the on-target. Purple triangles are rates from Ivanov et al.[42], when available at zero torque. **d** Microscopic free-energy landscape for an off-target with a mismatch at position 3 (blue), together with the on-target free-energy landscape (pink). Red arrow indicates the free-energy penalty $\delta\epsilon_3$ at the mismatch, and black arrow indicates the resulting shift in barrier height. **e** The calculated coarse-grained forward and backward rates on an off-target with a mismatch at position 3. Orange arrow highlights the rate that changed considerably compared to on-target. Purple triangles are rates from Ivanov et al.[42], when available at zero torque. **f** Microscopic free-energy landscape for an off-target with a mismatch at position 15 (blue), together with the on-target free-energy landscape (pink). Red arrow indicates free-energy penalty $\delta\epsilon_{15}$ at the mismatch, and black arrow indicates the resulting shift in barrier height. **g** The calculated coarse-grained forward and backward rates on an off-target with a mismatch at position 15. Orange arrow highlights the rate that changed considerably compared to on-target. In Fig. 5c, e, and g, central line represents the median, the box plots represent the interquartile range, and whiskers represent the full range among our 33 near equivalent solutions.

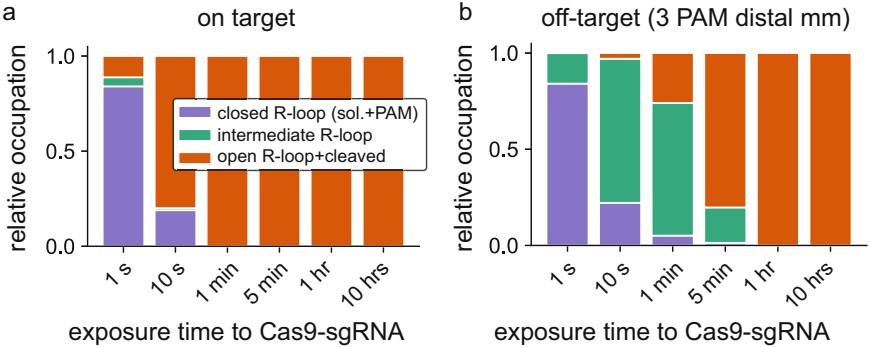

**Fig. 6 Dynamics among metastable states resemble structural dynamics. a** Time-resolved relative occupancy for the on-target among the closed R-loop state (solution and PAM bound), the intermediate R-loop state, and the open R-loop and cleaved state (c.f. Fig. 2d of Dagdas et al.[51]); **b** Relative occupancy at different time points for an off-target with the last 3 PAM distal base pairs mismatched (c.f. Fig. 2f of Dagdas et al.[51]).

(CFD) score[12], since it is a much-used tool for off-target classification.

To compare our model against the three selected off-target classifiers, we choose to rank all genomic sites based on cleavage activity in the low enzyme-concentration limit (see Methods). We make our comparison over all canonical PAM sites in the human genome. True positive off-targets are collected from sequencing-based cleavage experiments that used industry-standard sgRNAs

and reported multiple off-target cleavage sites[35–38,40,41,62] (Supplementary Table 1). We tested how well our kinetic model's ranking of activity compares to that of the CFD score[12], CRISPRoff[28], and uCRISPR[24]. For each sgRNA, we separately tested the models by using the union (sites found in any experiment) and intersection (sites found in every experiment) sets of the reported off-target sites as true positives. We perform precision-recall (PR) analysis (Supplementary Fig. 3) rather than

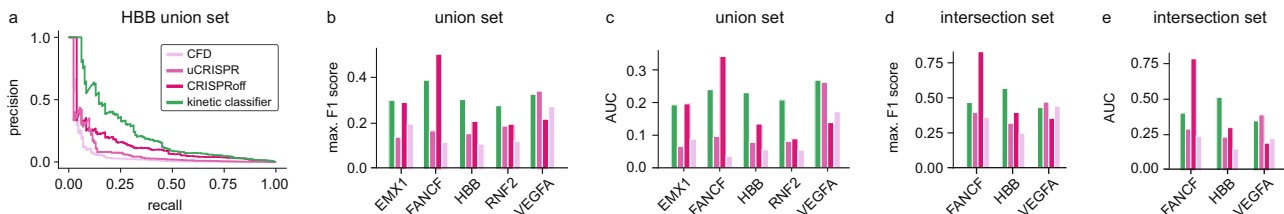

**Fig. 7 Genome-wide off-target classification. a** PR curves on the HBB gene using the CFD score (light purple), uCRISPR score (purple), CRISPRoff (dark purple), and our kinetic classifier (green). The precision and recall is calculated over all targets in the genome with a canonical PAM site, taking all experimentally validated off-targets as true positives. **b)** max. F1 scores for target sites EMX1, FANCF, HBB, RNF2 and VEGFA site 1 using all experimentally identified off-targets as true positives (union set) (Supplementary Fig. 3). **c** AUC scores for the same target sites and true positives as in **Fig. 7b. d** max. F1 scores using off-targets identified in all experiments as true positives (intersection set) (Supplementary Fig. 3). **e** AUC scores for the same target sites and true positives as in **Fig. 7d**. Matching the models pairwise we can determine which model performs best overall. Using max. F1 scores to count wins on union sets: kinetic:uCRISPR = 4:1; kinetic:CFD = 5:0; kinetic:CRISPRoff = 4:1. Using AUC scores to count wins on union sets: uCRISPR = 5:0; kinetic:CFD = 5:0; kinetic:CRISPRoff = 3:2. Using max. F1 scores to count wins on intersection sets: kinetic:uCRISPR = 2:1; kinetic:CFD = 2:1; kinetic:CRISPRoff = 2:1. Using AUC to count wins on intersection sets: uCRISPR = 2:1; kinetic:uCFD = 3:0; kinetic:CRISPRoff = 2:1. The kinetic classifier wins every pairwise matchup irrespective of if we use max. F1 or AUC scores, on both union and intersection sets.

using receiver-operator characteristics (Supplementary Fig. 4) since the datasets are highly unbalanced, with many more true negatives than true positives.

Figure 7a shows the PR curve when models are tested against the union of all reported off-targets while targeting the HBB gene. As the threshold for what is judged a strong off-target is swept, PR curves display the fraction of predicted off-targets that are found experimentally (precision) against the fraction of experimentally found off-targets that are also predicted (recall). Our kinetic classifier typically produces higher precision for all recalls, outperforming the other classifying schemes for the union set on the HBB gene. More importantly, the kinetic classifier also outperforms the leading off-target predictors for highly-mismatched genomic off-targets of other sgRNAs: performing best on the majority of targets in every pairwise matchup on both union (Fig. 7b, c) and intersection (Fig. 7d, e) sets, and irrespectively of if max. F1 or area under the curve (AUC) scores are used.

## Discussion
Training our model (Fig. 1) of SpCas9 target activity on moderately mismatched targets, we extract the physical parameters (Fig. 4) that control activity on any target (Figs. 2 and 3). Going beyond present-day binary off-target classification schemes, we quantitatively predict cleavage and binding activity as a function of both time and SpCas9-sgRNA concentration.

We show that SpCas9's targeting reaction contain an intermediate R-loop state, with both position and conversion rates that agree with single-molecule experiments[42] (Fig. 5). Mismatches affect the dynamics of the R-loop states (Fig. 6) in a manner similarity to how they affect the configurational states of SpCas9's nuclease domains[42,51,53]. Based on this, we lend support to the notion that R-loop formation is tightly coupled to protein conformation—pointing toward the relevant structure-function relation for the most important RNA-guided nuclease in use today.

Though our model captures the abundant low-activity off-targets that are discarded by binary classifiers, we sought to demonstrate the general utility of kinetic modelling by reducing our quantitative activity predictor to a binary classification tool. The resulting kinetic classifier outperforms established state-of-the-art classification tools on canonical PAM sites in the human genome (Fig. 7).

In a recent study, Jost et al.[5] demonstrated that a series of mismatched guides can be used to titrate gene expression using CRISPRa/CRISPRi. Wildtype SpCas9 can also be (effectively)

inactivated with PAM-distal mismatches in the guide[63]. Our model can guide such titration of SpCas9-sgRNA inactivation by careful placement of mismatches. Our approach can also be used to calculate the total off-target activity over a genome, and so inform the design of sgRNAs for novel gene targets.

For simplicity and robustness, we built our model to exclude mismatch type parameters. This allows for extensive training using datasets based on a single guide sequence and off-target DNAs containing up to two mismatches. The limited set of adjustable model parameters (44 in total) and efficient data usage (422 data points used for training) does not seem to limit the model's applicability (Figs. 2, 3, 7). The success of our low-complexity model strongly suggest that the path to increased predictive power and therapeutic relevance runs through bottom-up modelling of RNA-guided nucleases in kinetic terms.

Taken together, we have shown that our mechanistic and kinetic model gives biophysical insight and quantitative predictive power far beyond the training sets. This predictive power is only expected to increase when including sequence features and allowing for alternative PAM sequences in future modelling efforts. SpCas9 is only one of many RNA-guided nucleases with biotechnological applications, and other CRISPR associated nucleases (such as Cas12a, Cas13 and Cas14) offer a diversified genome-engineering toolkit[15,64–69]. These nucleases can all be characterized with our approach, and it will be especially interesting to compare the free-energy landscape of our SpCas9 benchmark to that of engineered[41,54,70] and natural (e.g. N. meningitides Cas9[71]) high-fidelity Cas9 variants.

## Methods
**Modelling of the (d)Cas9 targeting reaction.** We consider a single DNA target sequence with a PAM, in contact with (d)Cas9-sgRNA in solution at fixed concentration (Fig. 1a). (d)Cas9-sgRNA binding to the PAM site is assumed to be first order,

$$k_{on} = k_{on}^{ref}[Cas9 - sgRNA]$$

where [Cas9-sgRNA] is the concentration of active complexes relative to some reference concentration (we use 1 nM). Binding is followed by a Cas9-mediated strand exchange reaction between sgRNA and the DNA. Once a 20 bp hybrid is formed, Cas9 can cleave the DNA, while dCas9 cannot. We model the targeting recognition as a stochastic hopping process along a sequence of states: target unbound ($n = -1$), PAM bound ($n = 0$), and strand exchange ($n = 1, 2, \ldots, 20$). We use the column vector $\mathbf{P}(t) = (P_{-1}(t), \ldots, P_{20}(t))^T$ to represent the probabilities of being in the various states at time $t$. The evolution of probabilities is captured by the Master Equation

$$\partial_t \mathbf{P}(t) = \mathbf{K} \cdot \mathbf{P}(t),$$

where $\mathbf{K}$ is a tri-diagonal rate matrix. Letting $k_n^f$ be the forward ($n \rightarrow n + 1$) transition rate, $k_n^b$ to be the backward ($n \rightarrow n - 1$) transition rate (Fig. 1a), and defining $k_{-1}^b = 0$,

we can give the elements of **K** as

$$\mathbf{K}_{nm} = \begin{cases} k_{n-1}^{\mathrm{f}} & m = n-1 \\ -(k_n^{\mathrm{f}} + k_n^{\mathrm{b}}) & m = n \\ k_{n+1}^{\mathrm{b}} & m = n+1 \\ 0 & |n-m| \geq 2 \end{cases}.$$

The Master Equation has the formal solution

$$\mathbf{P}(t) = \exp(\mathbf{K}t) \cdot \mathbf{P}(0)$$

which can be computed numerically, given any set of rates **K** and initial probabilities **P**(0). The above expression, with initial probabilities and rates adjusted to experimental conditions (see below), allows us to capture the full time-dependent evolution of the targeting reaction in quantitative terms.

**Parameter reduction.** Based on the mechanistic-model assumption 1, we average the data over mismatch types (see below), and only keep track of if there is a match or a mismatch at every position. Model assumption 3 means that the model of dCas9 is the same as for Cas9, but with $k_{20}^{\mathrm{f}} = 0$. Model assumption 4 implies that $k_0^{\mathrm{f}} = k_1^{\mathrm{f}} = \ldots = k_{19}^{\mathrm{f}} \equiv k_{\mathrm{f}}$. To see the implications of model assumption 2, we move to a description in terms of free energies.

Denote the free energy of any state $n$ with $F_n$, and imagine that states $n$ and $n-1$ are allowed to mutually equilibrate. Equilibration means that the relative occupancy is described by Boltzmann weights and that there are no net probability currents between the states

$$\frac{P_{n-1}^{\mathrm{EQ}}}{P_n^{\mathrm{EQ}}} = \frac{\exp\left(-\frac{F_{n-1}}{k_{\mathrm{B}}T}\right)}{\exp\left(-\frac{F_n}{k_{\mathrm{B}}T}\right)}, \; P_{n-1}^{\mathrm{EQ}} k_{n-1}^{\mathrm{f}} = P_n^{\mathrm{EQ}} k_n^{\mathrm{b}}.$$

The above relationships tie rates to free-energy differences through

$$\Delta F_n = F_n - F_{n-1} = k_{\mathrm{B}}T \ln\left(\frac{k_n^{\mathrm{b}}}{k_{n-1}^{\mathrm{f}}}\right).$$

Using $n = -1$ as the free-energy reference ($F_{-1} = 0 \, k_{\mathrm{B}}T$), the assumption that binding is first-order implies

$$F_0 = F_0^{\mathrm{ref}} - k_{\mathrm{B}}T \ln([\mathrm{Cas9} - \mathrm{sgRNA}]).$$

Here $F_0^{\mathrm{ref}}$ is the free energy of the PAM bound state at the reference concentration (1 nM). Mechanistic-model assumption 2 now implies that $\Delta F_{1 \leq n \leq 20}$ only depends on if there is a mismatch at position $n$ or not, and we can write

$$\Delta F_n = \begin{cases} \epsilon_n, & \text{match} \\ \epsilon_n + \delta\epsilon_n & \text{mismatch} \end{cases}, \; n = 1, \ldots 20.$$

Here $\epsilon_n$ is the free-energy increase when extending the hybrid from length $n-1$ to length $n$ if the $n$:th hybrid bp is correctly matched, and $\delta\epsilon_n$ is the additional energy needed when the bp is incorrectly matched. We can write the backward transition rates as

$$k_n^{\mathrm{b}} = \begin{cases} k_{\mathrm{on}}^{\mathrm{ref}} \exp(\frac{F_0^{\mathrm{ref}}}{k_{\mathrm{B}}T}), & n = 0, \\ k_{\mathrm{f}} \exp(\frac{\Delta F_n}{k_{\mathrm{B}}T}), & n = 1, \ldots, 20. \end{cases}$$

The model is now parameterized it in terms of 41 free energies ($F_0^{\mathrm{ref}}, \epsilon_1, \ldots, \epsilon_{20}, \delta\epsilon_1, \ldots, \delta\epsilon_{20}$) and three forward rates ($k_{\mathrm{on}}^{\mathrm{ref}}, k_{\mathrm{f}},$ and $k_{\mathrm{cat}}$).

**Predicting NucleaSeq cleavage rates.** To produce predications for training and validation, we model experimental setups. To model NucleaSeq data[15], we use the solution to the Master Equation to calculate the expected cleaved fraction at any complementarity pattern. NucleaSeq is performed by exposing targets to saturating concentrations of Cas9-sgRNA, which we model by setting $F_0 = -1000 k_{\mathrm{B}}T$ and taking $P_{-1}(0) = 1$, $P_{0 \leq n \leq 20}(0) = 0$ as initial condition. As done in the original experiment, we record the fraction of DNA that remains uncleaved ($\sum_{n=-1}^{20} P_n(t)$) at the time points $t = 0$ s, 12 s, 60 s, 180 s, 600 s, 1800 s, 6000 s, 18000 s, and 60000 s, and fit-out a single effective cleavage rate $k_{\mathrm{clv}}^{\mathrm{eff}}$. There is no a priori reason for the uncleaved fraction to follow an exponential decay, but as long as we follow the experimental data-analysis protocol we can use the effective cleavage rates to train and validate our model.

**Predicting CHAMP association constants.** We model the CHAMP experiments[15,31] by calculating the bound fraction ($\sum_{n=0}^{20} P_n(t)$) of dCas9-sgRNA after 10 min at concentrations 0.1 nM, 0.3 nM, 1 nM, 3 nM, 10 nM, 30 nM, 100 nM and 300 nM, starting with the probabilities $P_{-1}(0) = 1$, $P_{0 \leq n \leq 20}(0) = 0$. We use the equilibrium binding fraction

$$P_{\mathrm{bnd}}^{\mathrm{EQ}} = \frac{[\mathrm{Cas9} - \mathrm{sgRNA}]}{[\mathrm{Cas9} - \mathrm{sgRNA}] + 1/K_{\mathrm{A}}^{\mathrm{eff}}}$$

to fit out an effective association constant $K_{\mathrm{A}}^{\mathrm{eff}}$. Again, there is no a priori reason to believe that this non-equilibrium system will equilibrate within

10 min, but as long as we follow the experimental data-analysis protocol we can use $K_{\mathrm{A}}^{\mathrm{eff}}$ for training and validation.

**Predicting HiTS-FLIP association rates.** To predict measured association rates in the HiTS-FLIP experiment[11], we assume the recorded fluorescence signal to be proportional to our calculated bound fraction of dCas9-sgRNA, when starting with the probabilities $P_{-1}(0) = 1$, $P_{0 \leq n \leq 20}(0) = 0$. Following the experiments we use linear regression to extract an effective association rate by fitting a straight line to the bound fraction at time points 500 s, 1000 s and 1500 s.

**Predicting HiTS-FLIP dissociation rates.** To predict measured dissociation rates in the HiTS-FLIP experimen[11], we again compared the fluorescence signal to our calculated bound fraction of dCas9, starting with the probabilities $P_{-1}(0) = 1$, $P_{0 \leq n \leq 20}(0) = 0$. We let the protein associate at saturating concentrations for 12 h, and record the resulting occupational probabilities. We then use these probabilities as new initial probabilities, while also letting $k_{\mathrm{on}} = 0$ ($[\mathrm{Cas9} - \mathrm{sgRNA}] = 0$) in **K**, before further evolving the system. This allows us to model complex dissociation in the presence of a high concentration of competitor on-targets in solution. Following the experiments, we fit an exponential decay to our predictions at time-points 500 s, 1000 s, and 1500 s.

**Averaging over mismatch types.** Our model does not account for mismatch types, and for training we need to average over all experimentally measured mismatch sequences $s$ consistent with a mismatch pattern $p$. We expect rates to be proportional to exponentiated transition-state free energies, and association constants to be controlled by exponentiated binding free energies. We therefore choose to perform our mismatch-type averages over the logarithm of rates and association constants, bringing these averages close to averages of energies. For measured quantities $m = k_{\mathrm{clv}}^{\mathrm{eff}}$ or $K_{\mathrm{A}}^{\mathrm{ref}}$, we chose a weighted mismatch-type average

$$\langle \log_{10} m^* \rangle_p = \sum_{s \in \left( \substack{\text{sequences with} \\ \text{mm pattern } p} \right)} W_s \log_{10} m_s^*.$$

Here $m_s^*$ is the measured value for target sequences $s$. We take the weights to be given by

$$W_s = \frac{1/\delta(\log_{10} m_s^*)^2}{\sum_{\sigma \in \left( \substack{\text{sequences with} \\ \text{mm pattern } p} \right)} 1/\delta(\log_{10} m_\sigma^*)^2}.$$

Here $\delta(\log_{10} m_s^*)$ is the experimental error for the logarithm of the measurement at a particular sequence $s$. This choice of weights minimizes the error-normalized square deviation on the sequence resolved data, if we have complete freedom to set the average for each mismatch pattern. Our model is more constrained then this, but with this weighing our model could—at least in principle—give the best possible approximation of the sequence resolved data. The squared error in the mismatch-type average can be calculated as

$$\delta.$$

**Cost function.** We look to simultaneously optimize our predictions of both effective cleavage rates from NucleaSeq ($k_{\mathrm{clv}}^{\mathrm{eff}}$) and effective dissociation constants from CHAMP ($K_{\mathrm{A}}^{\mathrm{ref}}$). We combine the cost from each experiment

$$\chi^2 = \chi_{k_{\mathrm{clv}}^{\mathrm{eff}}}^2 + \chi_{K_{\mathrm{A}}^{\mathrm{ref}}}^2$$

by summing log deviations

$$\chi_m^2 = \sum_{p \in \left( \substack{\text{all mm patters} \\ \text{used for training}} \right)} w_p^m (\log_{10}(m_p) - \langle \log_{10} m^* \rangle_p)^2.$$

In the above $m_p$ represent the model prediction for the average measured quantity at mismatch pattern $p$. The weights $w_p^m$ are chosen so the error-weighted contribution from the on-target, the 20 singly mismatched off-targets, and the $20 \cdot 19/2 = 190$ doubly mismatched off-targets are weighted equally as groups

$$w_p^m = \frac{1}{\delta\langle \log_{10} m^* \rangle_p^2} \cdot \begin{cases} 1, & p = \text{on target} \\ 1/20, & p \in \text{single mm} \\ 1/190, & p \in \text{double mm}. \end{cases}$$

**Simulated annealing.** The Simulated Annealing algorithm[59] is commonly used for high-dimensional optimization problems. We optimize with respect to model parameters $F_0^{\mathrm{ref}}, \epsilon_1, \ldots, \epsilon_{20}, \delta\epsilon_1, \ldots, \delta\epsilon_{20}$, $\log_{10}(k_{\mathrm{on}}^{\mathrm{ref}}/\mathrm{s})$, $\log_{10}(k_{\mathrm{f}}/\mathrm{s})$, and $\log_{10}(k_{\mathrm{cat}}/\mathrm{s})$. Trial moves are generated by adding a uniform noise of magnitude $\alpha$ to the present value of each model parameter. The process is initiated with a noise strength $\alpha = 0.1$. In the initiation cycle the temperature is adjusted until we have an acceptance fraction of 40–60% over 1000 trial moves, based on the Metropolis condition. After this initial

cycle, the temperatures follow an exponential cooling scheme with a 1% cooling rate ($T_{k+1} = 0.99 T_k$). At every temperature, we adjust the noise strength $\alpha$ until an acceptance fraction of 40–60% is reached over 1000 trial moves. Once the desired acceptance fraction is reached, an additional 1000 trial moves are performed to allow the system relax before the next cooling step. Once the temperature has dropped to one percent of its initial value we, apply the stop condition

$$|\bar{\chi}_k^2 - \bar{\chi}_{k-1}^2| \le 10^{-5} \bar{\chi}_{k-1}^2.$$

In the above, $\bar{\chi}_k^2$ denotes our cost function averaged over the last 1000 trial moves performed at temperature $T_k$. The results of this optimization is shown in Fig. 4.

**Calculating coarse-grained transition rates**. First we find the intermediate state on every possible target. As the central-local minimum in free energy (Fig. 4a) can be slightly displaced by mismatches on off-targets, we seek the free-energy minimum $n_I$ between R-loop state 7 and 13 for every target. To calculate the effective rates of the coarse-grained model in Fig. 5a, we consider the first passage between metastable states. Take for example the passage from the PAM-bound state ($n = 0$) to the intermediate state ($n = n_I$) on a specific target. To calculate the associated first-passage time, we truncate the full system to only include states $n = 0, \dots, n_I - 1$. We use the rate matrix $\mathbf{K}_{PI}$ with elements

$$(\mathbf{K}_{PI})_{nm} = \mathbf{K}_{nm}, \ 0 \le n, m \le n_I - 1$$

and $k_0^b = 0$. With the initial state $\mathbf{P}_{PI}(0) = (1, 0, \dots, 0)^T$ we solve the Master Equation, and calculate the first-passage time distribution as

$$\Psi_{PI}(t) = -(1, \dots, 1) \cdot \partial_t \mathbf{P}_{PI}(t).$$

The effective transition rate $k_{PI}$ is the inverse of the average first-passage time $\tau_{PI}$, which can be calculated as

$$\tau_{PI} = \int_0^\infty dt \, t \Psi_{PI}(t) = (1, \dots, 1) \cdot \mathbf{K}_{PI}^{-1} \cdot \mathbf{P}_{PI}(0).$$

The same process was used to calculate all other rates of directly transitioning between meta-stable states, repeated over every target sequence.

**Constructing a binary off-target predictor**. We rank all canonical PAM sites in the human genome according to their relative cleavage rate in the low concentration limit. In this limit, the cleavage rate is given by the PAM binding rate times the probability to cleave once the PAM site is bound. As the PAM binding rate is not expected to depend on the sgRNA sequence $s$, we can rank our off-targets based on the cleavage probability once bound[30],

$$P_{PAM \to clv}(s) = \frac{k_{cat} e^{\frac{F_{-1}(p(s))}{k_B T}}}{k_{cat} \sum_{n=0}^{19} e^{\frac{F_n(p(s))}{k_B T}} + k_f e^{\frac{F_{20}(p(s))}{k_B T}}}.$$

Here $p(s)$ is the mismatch pattern of sequence $s$.

**Statistics & Reproducibility**. Only experimental data giving physical positive values for mismatch-averaged rates and association constants were included in the correlation analysis. See Supplementary Data 1.

**Reporting Summary**. Further information on research design is available in the Nature Research Reporting Summary linked to this article.

## Data availability

The data supporting the findings of this study are available from the corresponding authors upon reasonable request. Mismatch averaged experimental data used for training and validation (Figs. 2 and 3), estimated microscopic parameters (Fig. 4), and genome wide off-target classification evaluation (Fig. 7b–e), are all provided as Supplementary Data 1.

## Code availability

The code enabling quantitative off-target activity prediction for any guide-target pair is available on our GitLab page (https://gitlab.tudelft.nl/depken_group/SpCas9_kinetic _model_dashboard). There you will also find a small dashboard application, allowing time resolved activity predictions given a particular sequence and enzyme concentration. A clone of the repository at publication is also permanently available at https://doi.org/10.5281/ zenodo.5790798. The purpose made optimization code will be made available upon request.

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

## Acknowledgements

We would like to thank Kristian Blom, Diewertje Dekker, and Sonny de Jong for valuable discussions and/or their help during the project. We also thank the members of the Chirlmin Joo lab and Stan Brouns lab for valuable discussions. We thank Evan Boyle for sharing data and answering all our questions. This work was supported by: Netherlands Organization for Scientific Research (NWO) (FOM-140), B.E.M.; Zwaartekracht Nano-Front, NWO M.K.; Parents in KIND program, The Kavli Institute of Nanoscience Delft/ the Department of Bionanoscience at TU Delft/through a Spinoza Prize awarded to M. Dogterom, M.D.; University of Texas College of Natural Sciences Catalyst award and the Welch Foundation (F-1808) I.J.F.; U.S. National Institute of Health (R01GM124141, F32AG053051) I.J.F. and S.K.J.

## Author contributions

B.E.M. and M.K.: Designed and performed the research, and wrote the manuscript K.v.dS. and C.v.dS.: Performed the research. S.K.J.: Provided data, and wrote manuscript J.H.: Provided data, and wrote manuscript I.J.F.: Provided data, and wrote manuscript M.D.: Conceived of the project, designed the research, and wrote the manuscript.

## Competing interests

The authors declare no competing interests.
