## [Peer Review File · Nature Communications]

Title: A kinetic model predicts SpCas9 activity, improves off-target classification, and reveals the physical basis of targeting fidelityREVIEWER COMMENTS

Reviewer #1 (Remarks to the Author):

Eslami-Mossallam et al. present the application of a kinetic model to predict off-targets for binding and cleavage by Cas9. They furthermore state that the model reveals the physical basis of SpCas9 fidelity. The model predictions are based on a very similar model that the Depken group introduced previously. The novelty in this study is that they train the model on available HITS-Flip data and apply it to predict off-targets for untrained data sets with high confidence. The significant achievement of this study is that their model outperforms other state-of-the-art off-target prediction tools, which definitely has its merits. I am more skeptical about the notation that the model reveals the physical basis of Cas9 fidelity. The obtained results should also be better compared and discussed to other studies.

1) Generally the model predictions with respect to high-throughput sequencing data appear to be rather solid. Somewhat a caveat is however that the forward rate k_f is very ill-defined and stretches apparently over almost 3 orders of magnitude, which affects also the energy landscape of the R-loop progression. I strongly suggest to use the available dissociation data of Boyle et al. to parametrize k_f better. This data should also provide a consistency check for the established model.

2) The model does not include any sequence-dependence of the mismatch penalties and was trained by averaging rates/affinity constants for a limited number of different mismatches. To what extent does the roughness in the energy landscape (Fig. 2a) reflect the scatter from the sequence-dependence of the local means of the training data? Taking the obtained energy landscape serious there should be another intermediate around position 15. Is this really occurring or just an artifact of the model parametrization? If so the error bars on the energy landscape would be quite high. The authors should discuss this more detailed and/or provide realistic error bars for their landscapes.

3) The authors make a strong statement about the novel physical basis that the model provides which occupies a big part of the manuscript. I am less convinced about the merits of the manuscript in this aspect. The identification of the 3 different intermediates has already been achieved previously, e.g. in single molecule FRET experiments but also in qualitative discussions of sequencing data. The authors simply adapt this notation. Based on their landscape I would however rather introduce an additional state around position 15. I suggest a more honest discussion of the achievements of the current study. I am not fully sure how much Figs. 3c-j contribute to the manuscript, since they are just inferred theoretically without comparison to data.

4) Regarding the data of Dagdas et al. a more quantitative rather than a qualitative comparison would be helpful, particularly if shown at this detail. I am a bit surprised why for 3 and more mismatches the C state can be occupied et al. since the mismatch penalties should essentially remove any local energy minimum at this position.

5) The authors should certainly compare their model predictions with the recent publication of the

Bryant and Doudna labs (Ivanov et al. PNAS). These authors measure actually directly the DNA untwisting and thus the positions of the I and the C states as well as the transition rates between the different states. This data would in my opinion be an even better reference for judging the goodness of the energy landscape and rate predictions.

6) The authors should more clearly point out that their model has been essentially introduced in a previous publication by Klein et al.. They should also give credit to studies that model DNA strand displacement reactions in DNA nanotechnology (without and with mismatches) with essentially the same theoretical approach.

7) Please explain on page 4 the notation "we confirm partial equilibration of the system", which is not clear to me.

Reviewer #2 (Remarks to the Author):

Eslami-Mossallam presents an off-target analysis and prediction in the manuscript "A kinetic model improves off-target predictions and reveals the physical basis of SpCas9 fidelity". The work presents an interesting kinetic approach model CRISPR off-targeting.

(1) The article close resemble the authors previous published work (reference 28 in the manuscript: "Klein, M., Eslami-Mossallam, B., Arroyo, D. G. & Depken, M. Hybridization Kinetics Explains CRISPR-Cas Off-Targeting Rules. Cell Rep. 22, (2018)") and it is not so clear what is new compared to the previous paper. The authors themselves do not for some reason explicitly point to this. The authors should clearly indicate what are the differences between the model they present in the manuscript and their previously published work (reference 28 in the manuscript). In particular, they should make a comparison between the figure 1A from their previous work (ref. 28) and figure 1A in the present manuscript, to point out the novelties of the current model.

(2) The Doench and Zhang models are in the Introduction referred to as "the two best-performing genomic off-target prediction tools used today". However, the references used to justify this statement (12, 24, 42) are not sufficient: the independent benchmark of Haeussler et al. Genome Biology (2016) does not include the Zhang model as well as models developed after 2016. Newer work includes Listgarten et al. Nat Biomed Eng (2018) Elevation (update of the CFD method) and Alkan et al. Genome Biology (2018) presenting CRISPRoff which outperform both Elevation and CFD. It is also worth mentioning that CRISPRoff makes use of the Boyle data in its energy model in contrast to the version used in uCRISPR that was published later.

(3) The AUC plots on the same methods seem very different e.g. on CFD compared to what has been published elsewhere, e.g. Haeussler et al., and Alkan et al.

(4) Focusing on the canonical PAM is not sufficient. Cleavage at alternative PAMs cannot be ignored in the context of off-target predictions, as false negatives are harmful for practical/therapeutic application. The authors should take this aspect into account in their analysis.

(5) In contrast with other tools that precede off-targets evaluation with a genome-wide detection phase, the method presented does not predict off-targets, but rather evaluates them given off-target loci. The authors need to clarify this, as the method cannot be independently applied without making use of external tools for off-targets detection.

(6) The precision-recall curve of EMX1 should be substituted with ROC-AUCs, one for each sgRNA, as the F1 score already summarizes precision and recall, which are anyway fully reported in the supplement. The ROC-AUC score should also numerically appear in the text.

(7) The comparison to other models lacks some essential information, which the authors should address:

(i) What are the datasets used to train the CFD and uCRISPR models? If the models are trained on different datasets (eg. of very diverse sizes), how can they be compared in a fair way?

(ii) What is the level of similarity between the data used to generate the model described in the manuscript (as well as the alternative tools being evaluated) and the test sets?

(iii) How do these results fit to previous benchmarks on the same/similar datasets?

(iv) CFD and uCRISPR are not the most recent models available in the field, and no independent benchmark comparing their performances to those of the latest available models is cited (see comments above).

(8) According to the Results section, the method performs well on datasets containing off-targets with more than 2 mismatches. However, in the Supplementary methods section "Parameter estimation" the limit to two mismatches is justified by the fact that "the cleavage activity is much lower for three mismatches and above". (i) To prove that the model is generalizable to off-targets with more than 2 mismatches, and that it does not simply predict them all as negatives, the results/supplement should report the distribution of the predicted cleavage values produced by the method being presented for these off-targets as well as performances evaluated specifically on this subset of the data. (ii) A range of the other off-target methods use up to 6 mismatches as also reported in the literature (Kleinstiver et al. Nat Biotech 2015) and the authors furthermore include analysis of up to 6 mismatches.

(9) A section describing the databases used in the study and how the input data looks like is necessary to interpret their usage in the model. The authors should include this.

(10) A minimum version of the code used in this study should be provided. It was not included in the submission.

Minor

(11) The scale and unit of measure should be reported in each figure (eg. Fig 1B, 2A, 2B, 3B...).

(12) The legend and caption of figure 1B are incomplete: are the red dashed lines representing the mismatch at pos. 3 and 15, and not the blue bullets? Also, the abbreviation "sol." is not described (it is later on explained in the caption of Figure 2).

(13) It is a good practice to report the number of data points used to construct box plots, or to scatter the data in addition to the boxes (eg. Fig 3).

(14) Fig. 4B and C are in the wrong order. The 4 PAM distal mm figure should be 4C, not 4B, according to the caption (and vice versa for 3 PAM distal mm).

(15) The TPR-FPR plots in Supplementary Figure 5 have different scales: this makes them hard to read. Also, a 0-1 scale should be used on both X and Y axis, avoiding unnecessary exponential notation.

Reviewer #3 (Remarks to the Author):

In this very interesting and well-written manuscript, Dr. Depken and his co-authors have presented a detailed kinetic model of the function of the SpCas9 genome editing system, focusing in particular on how this system works to control the fidelity of target identification and cleavage. The major advance that this model makes over previous models -- in which equilibrium at each step is assumed -- is that here the authors take into account the fact that metastable intermediate states may well appear and may dominate important aspects of the function of the system.

The model is complicated, and presumably mechanistically constructed correctly on the basis of earlier literature in the field. The major steps involve initial binding of the spCas9 complex to a 3nt protospacer adjacent motif (PAM) in the genomic DNA, which results in local opening of the double-stranded DNA and the initiation of a R-loop (RNA-DNA hybrid), which either eventually expands to a 20 bp RNA-DNA hybrid loop that then triggers the double-stranded DNA cleavage event and genome editing, or -- at any step in the R-loop expansion -- can reverse and contract the loop, eventually resulting in dissociation without cleavage. Single bp mismatches within the R-loop provide finite barriers to further expansion and thus serve to kinetically control the fidelity of the system.

As a kinetic model this system, in principle, contains an unmanageable multiplicity of kinetic parameters, and so the authors are compelled to 'coarse-grain' their approach by making a number of simplifying assumptions, including, among others, that all bp mismatch types are treated as positionally important but otherwise the same, a mismatch only influences reversal of the mismatched pairing and all other kinetic parameters remain the same, and that hybrid-formation rates are the same, regardless of local

sequence or complementarity.

These assumptions are likely not true in detail, but they work and so obviously differences in these parameters don't significantly perturb the outcome. The authors then test their results against experimental datasets and show that the predictive power (in terms of cleavage infidelity) of their model is better than those that are currently used, which -- as indicated above -- assume equilibrium throughout and don't take metastable states into account.

I am not an expert on the detailed structural and functional literature of these genome-editing systems, and so assume that the authors have correctly represented the literature and tested their model against the available datasets testing for cleavage fidelity. On this basis the model the authors have set up is elegant and clearly leads to interesting results, including further insight into how these systems must function in mechanistic terms. On this basis I am pleased to state that in my opinion this work is likely to be of considerable significance in the field, and is thus very appropriate for publication, together with the very informative and well-written Supplementary Information (SI) section, in Nature Communications.

Review by Peter von Hippel

Rebuttal

We would like to thank all three reviewers for their careful reading of our manuscript and their thoughtful comments. The reviewers have highlighted many valid points of improvements, offered many great suggestions, and we feel that their broad expertise has allowed us to substantially improve our paper. Below you will find our point-by-point answer to the reviewers, as well as a detailed list of amendments made.

In his referee report, Prof. von Hippel very nicely explained how the reaction scheme in **Figure 1a** is only the beginning of the work, and the real modelling consists of coming up with physical rules to reduce the parameter space. As we were pressed by both the other reviewers to better clarify the novel modelling done here, we find it efficient to start by addressing the comments of Prof. von Hippel.

Below the reviewers comments are reproduced in *italics*, and we give our responses unitalicized. All figure/section references are referring to the numbering used in the present version of the manuscript. The line numbers refer to the main manuscript, which includes all the main figures and captions incorporated in the text, and all supplementary figures and captions at the end of the same document.

Reviewer 3:

In this very interesting and well-written manuscript, Dr. Depken and his co-authors have presented a detailed kinetic model of the function of the SpCas9 genome editing system, focusing in particular on how this system works to control the fidelity of target identification and cleavage. The major advance that this model makes over previous models -- in which equilibrium at each step is assumed -- is that here the authors take into account the fact that metastable intermediate states may well appear and may dominate important aspects of the function of the system.

The model is complicated, and presumably mechanistically constructed correctly on the basis of earlier literature in the field. The major steps involve initial binding of the spCas9 complex to a 3nt protospacer adjacent motif (PAM) in the genomic DNA, which results in local opening of the double-stranded DNA and the initiation of a R-loop (RNA-DNA hybrid), which either eventually expands to a 20 bp RNA-DNA hybrid loop that then triggers the double-stranded DNA cleavage event and genome editing, or -- at any step in the R-loop expansion -- can reverse and contract the loop, eventually resulting in dissociation without cleavage. Single bp mismatches within the R-loop provide finite barriers to further expansion and thus serve to kinetically control the fidelity of the system.

As a kinetic model this system, in principle, contains an unmanageable multiplicity of kinetic parameters, and so the authors are compelled to 'coarse-grain' their approach by making a number of simplifying assumptions, including, among others, that all bp mismatch types are treated as positionally important but otherwise the same, a mismatch only influences reversal of the mismatched pairing and all other kinetic parameters remain the same, and that hybrid-formation rates are the same, regardless of local sequence or complementarity.

These assumptions are likely not true in detail, but they work and so obviously differences in these

parameters don't significantly perturb the outcome. The authors then test their results against experimental datasets and show that the predictive power (in terms of cleavage infidelity) of their model is better than those that are currently used, which -- as indicated above -- assume equilibrium throughout and don't take metastable states into account.

*I am not an expert on the detailed structural and functional literature of these genome-editing systems, and so assume that the authors have correctly represented the literature and tested their model against the available datasets testing for cleavage fidelity. On this basis the model the authors have set up is elegant and clearly leads to interesting results, including further insight into how these systems must function in mechanistic terms. On this basis I am pleased to state that in my opinion this work is likely to be of considerable significance in the field, and is thus very appropriate for publication, together with the very informative and well-written Supplementary Information (SI) section, in Nature Communications.
Review by Peter von Hippel*

Response:

- A. We are very pleased with the positive evaluation, and especially with description of the mechanistic modelling aspect of our paper. Prof. von Hippel points to the importance of the supplementary information, and we have now further extended and improved it.

We have:

- reorganized and extended the SI to clearly address specific topics raised by the other reviewers, as detailed below.
- introduced detailed referencing to the supplementary section for easy access to specific topics highlighted in the main text.

Reviewer 1

Eslami-Mossallam et al. present the application of a kinetic model to predict off-targets for binding and cleavage by Cas9. They furthermore state that the model reveals the physical basis of SpCas9 fidelity. The model predictions are based on a very similar model that the Depken group introduced previously. The novelty in this study is that they train the model on available HITS-Flip data and apply it to predict off-targets for untrained data sets with high confidence. The significant achievement of this study is that their model outperforms other state-of-the-art off-target prediction tools, which definitely has its merits. I am more skeptical about the notation that the model reveals the physical basis of Cas9 fidelity. The obtained results should also be better compared and discussed to other studies.

Response:

- B. We are happy that the reviewer finds our study of merit with regards to the off-target predictions, but we hope that our detailed responses below will also successfully convince them that this is not the main merit of our study. As we see it, the merit is twofold: 1) Present methods look for strong off-targets, even though the notion of strong is ill defined in this highly variable and stochastic system. This casts doubt on the very utility of binary classification tools, and we overcome this by constructing a quantitative activity measure accounting for probabilities. We only reduce our model to a binary classification tool to show the modelling community what is possible when taking an approach based in non-equilibrium physics. 2) We characterize SpCas9 in physical/kinetic terms. Contrary to existing models, all our parameters have direct physical meaning, and are interpretable in terms of observables (e.g. a free-energy minima corresponding to an R-loop intermediate). Our general approach will offer important

insights when applied to high fidelity variants, and we here establish a benchmark by characterizing SpCas9.

It is also important to note that we do not train on HiTS-FLIP experiment as stated by the reviewer, but we do quantitatively predict them, with no extra fit parameters (Figure 2b and c).

1) Generally the model predictions with respect to high-throughput sequencing data appear to be rather solid. Somewhat a caveat is however that the forward rate k_f is very ill-defined and stretches apparently over almost 3 orders of magnitude, which affects also the energy landscape of the R-loop progression. I strongly suggest to use the available dissociation data of Boyle et al. to parametrize k_f better.

Response:

- C. The reviewers comment encouraged us look into this, and we believe we understand what was happening. It is easy for the optimization to get stuck in local minima where an inflated forward rate is compensated for by an increased barrier between metastable states in the on-target (see present **Supplementary Section S4.1**). This is likely why there were a few solutions, stuck with very high barriers in the on-target, compensated for by very high forward rates. Restricting the set of solutions to just those that lie within 15% of the best solution, the range of k_f now collapses (**Figure 3c**).

We have:

- included our predictions for association and dissociation in Boyle et al. (**Figure 2b and c**).
- amended the plots to show solutions within 15% of the best solution (33 in total) (**Figure 3**).

2) The model does not include any sequence-dependence of the mismatch penalties and was trained by averaging rates/affinity constants for a limited number of different mismatches. To what extent does the roughness in the energy landscape (Fig. 2a) reflect the scatter from the sequence-dependence of the local means of the training data?

Response:

- D. As the roughness in the barrier regions of the on-target landscape are not reproducible between optimization runs, we do not expect it to be generated by a reproducible effect like the guide sequence. This roughness instead results from the many barriers that give the same effective rates.

Still, we thank the reviewer for asking about this roughness, as it prompted us to look at the roughness in mismatch penalties (**Figure 3b**), which could be influenced by the effect the reviewer is pointing to; and, indeed, it seems highly reproducible. This roughness could certainly be due to the guide sequence, but could also be due to interactions with the protein. To determine the source, we would need to train on multiple guides. Though training on multiple guides is outside the scope of this paper, we take the fact that we outperform existing off-target classifiers over multiple guides as an indication that the roughness is dominated by protein interactions.

We have:

- more clearly explained the roughness of the barriers (line 157)
- highlighted the reproducibility of the mismatch penalty roughness in the text (line 179)

- changed the plot of mismatch penalties (**Figure 3b**) to show all solutions, to emphasise the reproducibility of the variations.

Taking the obtained energy landscape serious there should be another intermediate around position 15. Is this really occurring or just an artifact of the model parametrization? If so the error bars on the energy landscape would be quite high. The authors should discuss this more detailed and/or provide realistic error bars for their landscapes.

Response:

- E. It is true that the “error bars” are too high at position 15 to assign it as a metastable state. In the present **Figure 3a** we illustrate this by plotting all 33 solutions that we found to be within 15% of the best solution. The “cloud of solutions” at position 15 is indeed too wide to interpret 15 as a metastable state, and should be contrasted with the vanishing cloud e.g. at position 11.

We have:

- removed the whisker plots of **Figure 3** and show all solutions instead, as not insinuate that we have proper error bars taking into account e.g. experimental variations.

3) The authors make a strong statement about the novel physical basis that the model provides which occupies a big part of the manuscript. I am less convinced about the merits of the manuscript in this aspect. The identification of the 3 different intermediates has already been achieved previously, e.g. in single molecule FRET experiments but also in qualitative discussions of sequencing data. The authors simply adapt this notation.

Response:

- F. The reviewer here casts doubt on a central claim of our paper, so it is important that we are precise about why we respectfully disagree with the reviewer’s assessment.

Our claim to novelty was never that we were the first to find a metastable state, but that our model “[...] reveals the physical basis of targeting fidelity” (title, and line 50). The existence of a metastable intermediate state explains why Cas9 is a more promiscuous binder than cleaver, but it does not allow you to make any quantitative predictions about cleavage fidelity. In short, it does not explain the physical basis for fidelity, as the reviewer seems to imply.

Fidelity is a statement about relative activity, and to address this we need a physical model capturing the physics on all targets; not just a handful of constructs considered in single-molecule experiments. The novel quantitative results presented in **Figure 3a** and **b**, when combined with our mechanistic rules (line 74), provides such a model. Before this work, there was no physical understanding that allowed genome wide activity predictions, and the field has had to make do with various *ad hoc* ranking schemes for scoring off-targets.

The reviewers point about us just adopting the configurational states observed from FRET experiments is well taken though, and we have now softened our claims in this regard (line 243). Conveniently, the paper the reviewer later points to (Ivanov et al. PNAS) directly observes the R-loop states we infer, and we no longer depend on the analogy to structural states to contextualize our finding. We thank the reviewer for pointing us to this important paper.

Though a minor point, we would also like to point out that the previous identification of three metastable states (see e.g. Dagdas et al. PNAS or Ivanov et al. PNAS) does not differentiate between the solution and the PAM bound state, but simply refers to them collectively as the closed state. This grouping implicitly means that equilibration is assumed for PAM binding, which is not *a priori* guaranteed. We identify the solution state and PAM bound states as different, and thus are able to separate out the physics that are intrinsic to the molecule (corresponding to states PAM, R1, R2, ..., R20), and that which are set by the experimental conditions (the solution state). This separation allows us to train the Cas9 intrinsic part over several experiments, varying only the experimental conditions (e.g. Cas9 concentration). In this sense, we do characterize more states than previously done, and in doing so we dispense with an implicit assumption of binding equilibrium over all PAM sites, and can train our model over multiple experiments and experimental conditions.

We have:

- further clarified what our central claim is (line 50).
- introduced the metastable R-loop states we find as already directly observed (line 53)
- changed the name of the effective rate from the PAM bound state to the intermediate state from kCI to kPI. This to avoid confusion with the rates defined in Ivanov and Dagdas papers, which refer to the rate from the combined solution+PAM state to the intermediate state as kCI.
- We have removed the claim of correspondence between conformational and R-loop states from the abstract, and toned down the claims in main text (line 233, 243), and conclusions (line 302).
- made explicit that the connection between conformational and R-loop states was already suggested in Ivanov et al. (line 233)

Based on their landscape I would however rather introduce an additional state around position 15.

Response:

- G. This is not supported by the data, as explained in our **response E** above.

I suggest a more honest discussion of the achievements of the current study.

Response:

- H. We have done our best to keep the discussion honest, and we hope that our **response F** (and the corresponding amendments) helps clarify to the reviewer what our central claim is. We now reference (Ivanov et al. PNAS) throughout our manuscript, which should help make clear that our claim to novelty is not that we identify novel metastable R-loop states, but that we explain fidelity by establishing the quantitative free-energy landscape for every guide-target combination.

I am not fully sure how much Figs. 3c-j contribute to the manuscript, since they are just inferred theoretically without comparison to data.

Response:

- I. These energy landscapes are fitted out from our high-throughput data using our mechanistic model assumptions. These are quantitative versions of precisely the type of qualitative plots that have been used to discuss the handful of constructs examined in single-molecule experiments (see e.g. Ivanov, PNAS). In this context, we find that these free-energy landscapes

nicely illustrate the application of the targeting rules, and that the corresponding effective rates illustrate how the dynamics of the system is controlled.

The utility of these plots should now also be further increased, as we use them to compare the effective rates we predict with those observed by Ivanov et al., as suggested by the reviewer under point 5 below. Unfortunately, not many rates in the Ivanov paper are directly comparable, because many of the measurements are not available at zero torque (our situation).

We have:

- made a clearer connection to experiments by overlaying the available rates from Ivanov et al. as purple triangles in the rate plots of **Figure 4b-d**.
- Included a discussion of the quantitative correspondence between our model and the Ivanov et al. at line 216.

4) Regarding the data of Dagdas et al. a more quantitative rather than a qualitative comparison would be helpful, particularly if shown at this detail.

Response:

- J. As too few time points are reported in Dagdas et al. to make a meaningful quantitative comparison, we opted to qualitatively show the trends in a manner similar to what was done in the Dagdas paper. Upon reflection, we do agree though that the way we presented it was confusing, as it reverted back to the very detailed microscopic picture after we have introduced the metastable states.

We have:

- lessened the detail of **Figure 5**, as not to suggest it predicts more than it does.

I am a bit surprised why for 3 and more mismatches the C state can be occupied et al. since the mismatch penalties should essentially remove any local energy minimum at this position.

Response:

- K. We understand the reviewer's surprise, as we unfortunately mislabelled the "closed and cleaved state" as just the "closed" state in **Figure 5**. The cleaved state is absorbing, resulting in a gradual build-up of its occupation over time.

It should also be noted that Ivanov et al. and Dagdas et al. used a different notation for the states (our closed state was their open state, and our open state only referred to the PAM bound state). This was highly confusing, and we apologize for this. To be consistent with the existing literature we have now adopted their notation for the full hybrid state to be the open R-loop state.

We have:

- We have corrected the legends of **Figure 5** to "open state & cleaved".
- changed our notation from (open, intermediate, closed) to (PAM bound, intermediate, open) R-loop states throughout.

5) The authors should certainly compare their model predictions with the recent publication of the Bryant

and Doudna labs (Ivanov et al. PNAS). These authors measure actually directly the DNA untwisting and thus the positions of the I and the C states as well as the transition rates between the different states. This data would in my opinion be an even better reference for judging the goodness of the energy landscape and rate predictions.

Response:

- L. We thank the reviewer for pointing us to this important work! This work beautifully and directly observes the R-loop states that we infer from bulk experiments (see also our **response F**). We now show the measured rates from Ivanov et al. in **Figure 4** (when available at zero torque), and highlight that we predict the location of the intermediate state as well.

We have:

- included references and mentions of the Ivanov et al. paper throughout the manuscript.
- Added data from Ivanov et al. to **Figure 4**
- shortened our discussion of the Dagdas paper to make room for a discussion of the Ivanov paper.

6) The authors should more clearly point out that their model has been essentially introduced in a previous publication by Klein et al.. They should also give credit to studies that model DNA strand displacement reactions in DNA nanotechnology (without and with mismatches) with essentially the same theoretical approach.

Response:

- M. We agree that we could have better contrasted the present work more clearly with previous work by us and that in the strand-displacement literature. What sets our present study apart from our previous proof-of-principle study (Klein et al.) and the strand-replacement literature, is that we here allow the free-energy gains/losses of bond formation to be influenced by interactions with the protein. This is a very important difference, as without it one cannot capture the observed intermediate metastable state in the on-target reaction, and thus would have little hope of capturing the actual physics of the present problem.

We have:

- included an explanation of the difference between this model and the ones for strand displacement reactions in solution and Klein et al. (line 100)
- added references to the relevant strand-displacement models on line 104.

7) Please explain on page 4 the notation "we confirm partial equilibration of the system", which is not clear to me.

Response:

- N. We are sorry that we were not very clear on this point, and thank the reviewer for pointing this out. We confirmed partial equilibration (equilibration within individual metastable states) of the system by showing that the predictions are invariant under the change of forward rate $k_f \rightarrow Q k_f$ coupled to a change in barrier height $G \rightarrow G + \ln(Q)$ (**Supplementary Figure S2a-d**). Had the fast degrees of freedom not been equilibrated, changing the forward rate would change the systems approach to equilibrium, and thus the predictions of our model. As the predictions do not change when we perform this transformation, we conclude that the fast degrees of freedom are equilibrated and we have captured all metastable states.

We have:

- added a detailed description of this in **Supplementary Section S4**.

Reviewer 2

Eslami-Mossallam presents an off-target analysis and prediction in the manuscript “A kinetic model improves off-target predictions and reveals the physical basis of SpCas9 fidelity”. The work presents an interesting kinetic approach model CRISPR off-targeting.

Response:

- O. We thank the reviewer for the thorough consideration and commenting on our manuscript, and the interest shown in our kinetic approach!

(1) The article close resemble the authors previous published work (reference 28 in the manuscript: “Klein, M., Eslami-Mossallam, B., Arroyo, D. G. & Depken, M. Hybridization Kinetics Explains CRISPR-Cas Off-Targeting Rules. Cell Rep. 22, (2018)”) and it is not so clear what is new compared to the previous paper. The authors themselves do not for some reason explicitly point to this. The authors should clearly indicate what are the differences between the model they present in the manuscript and their previously published work (reference 28 in the manuscript). In particular, they should make a comparison between the figure 1A from their previous work (ref. 28) and figure 1A in the present manuscript, to point out the novelties of the current model.

Response:

- P. A similar point was brought up by Reviewer 1 (see **response M** and amendments done), and though there is a large difference between the models, we realize that we should have made the distinction much clearer in the paper. The reaction scheme in **Figure 1a** does not represent our modelling contribution, as this was accepted in the literature well before our previous study. Our model contribution is our modelling assumptions, which drastically reduces the number of free parameters and allows for parameter estimation.

We have:

- separated out our mechanistic modelling assumptions into a bullet list (line 74), to emphasise that this is where the modelling particular to this paper is introduced.

(2) The Doench and Zhang models are in the Introduction referred to as “the two best-performing genomic off-target prediction tools used today”. However, the references used to justify this statement (12, 24, 42) are not sufficient: the independent benchmark of Haeussler et al. Genome Biology (2016) does not include the Zhang model as well as models developed after 2016. Newer work includes Listgarten et al. Nat Biomed Eng (2018) Elevation (update of the CFD method) and Alkan et al. Genome Biology (2018) presenting CRISPROff which outperform both Elevation and CFD. It is also worth mentioning that CRISPROff makes use of the Boyle data in its energy model in contrast to the version used in uCRISPR that was published later.

Response:

- Q. We thank the reviewer for pointing us to these references! It is a good suggestion that we also compare our results to those of Listgarten et al. (2018), and Alkan et al. (2018). As Alkan et al.

show that they outperform Listgarten et al., we only add a direct comparison to Alkan et al., but include Listgarten et al. as a reference.

We have:

- included a comparison to Alkan et al. in main text and supplement, showing that we outcompete this model (**Figure 6, Supplementary Figure S4 and S5**).
- added a reference to Listgarten et al. (line 22)

(3) *The AUC plots on the same methods seem very different e.g. on CFD compared to what has been published elsewhere, e.g. Haeussler et al., and Alkan et al.*

Response:

- R. Other authors limit the considered data considerably by preselecting potential target sites. For example, the analysis done in Alkan *et al.* considers only up to 6 mismatches, and thus side-steps many potential off-targets. We saw no good *a priori* reason to limit the data sets like this, and decided to compare the models over all canonical PAM sites in the human genome (see also our **response S** below). The fact that we do not limit the search depth influences the True/False positive rates, and thus the ROC curves look different. Furthermore, Haeussler and in Alkan *et al* aggregate different guides in a single ROC curve, while we show an individual ROC curve for every guide.

(4) *Focusing on the canonical PAM is not sufficient. Cleavage at alternative PAMs cannot be ignored in the context of off-target predictions, as false negatives are harmful for practical/therapeutic application. The authors should take this aspect into account in their analysis.*

Response:

- S. It is true that before our approach gets a clinical application, non-canonical PAMs need to be considered, but so does sequence, chromatin context, and many other environmental factors. It is our feeling that the off-target prediction field often gets ahead of itself in claiming to have therapeutically useful results without accounting for many of the environmental factors present. The simple fact that concentration and exposure time can be used to control fidelity *in vivo* (Tycko, MolCell 2016), but is still not captured by present off-target classifiers, highlights this very point. Though our model now captures the effects of varying concentrations and exposure times, there is still some way to go before we will claim to have a therapeutically useful tool—including the modelling of the PAM interaction the reviewer points to.

Our aim with reducing our fully time-resolved activity predictor to a binary off-target classifier was not to offer an alternative binary off-target classifier; we believe such approaches to be problematic as they miss the vast number of low probability off-targets present (see also our **Response A** above). Instead, we use the comparison to binary off-target predictors to show that the kinetic nature of the problem is more important than many details others include—such as sequence. This fact is presently not appreciated by the modelling community, and by outperforming current predictors, we hope to make a strong case for kinetic modelling, and the future inclusion of the features so far omitted.

We have:

- made it clearer that we are only considering canonical PAM sites (line 264 and 315).

(5) In contrast with other tools that precede off-targets evaluation with a genome-wide detection phase, the method presented does not predict off-targets, but rather evaluates them given off-target loci. The authors need to clarify this, as the method cannot be independently applied without making use of external tools for off-targets detection.

Response:

T. We are quite puzzled by this comment, as we did independently apply our approach without the use of any external tools. Our parameterized model simply gives us the time-dependent cleavage/binding activity on any target, which we then used to rank *all* canonical PAM sites in the human genome. It is unclear to us why the reviewer makes this claim; if it was prompted by anything we wrote (or omitted to write), we would be happy to amend this.

(6) The precision-recall curve of EMX1 should be substituted with ROC-AUCs, one for each sgRNA, as the F1 score already summarizes precision and recall, which are anyway fully reported in the supplement. The ROC-AUC score should also numerically appear in the text.

Response:

U. As we do not artificially limit the sequence space we test over (see **response R**), our data sets have many more True negatives than True positives. For such unbalanced data sets, the ROC-AUCs would simply not be very informative: the areas under the curves will all be very close to the maximal value 1, independently of the model considered. To see the reason for this, note the log axis in **Supplementary Fig. S5** together with the fact that the TPR have always essentially reached 1 before a FPR of 10^{-3} . We opted for PR curves and max F1 scores instead, as they do not involve the number of True positives or True negatives, and so allows for a clear comparison also on unbalanced datasets.

(7) The comparison to other models lacks some essential information, which the authors should address: (i) What are the datasets used to train the CFD and uCRISPR models? If the models are trained on different datasets (eg. of very diverse sizes), how can they be compared in a fair way?

Response:

V. We did not succeed in communicating our intent here, and apologize for this. Our purpose with the test against other models (**Figure 6**) is to show that our approach outperforms the others on the entire human genome. Unfortunately, uCRISPR use some of the genome wide data for training, allowing us only to determine when other models beat uCRISPR, and not the other way around. As we still outperformed uCRISPR on four out of five targets, it is safe to say that we perform better, and we did not see the need to perform the more sophisticated analysis.

We have:

- added a statement in the caption of **Figure 6** regarding the undue advantage for uCRISPR in our comparison.

(ii) What is the level of similarity between the data used to generate the model described in the manuscript (as well as the alternative tools being evaluated) and the test sets?

Response:

W. The model we use is not generated from the data, but is based on structural and biochemical information. We do estimate the parameters from the data, but this is very different, and it

forms an important distinction between top-down modelling and our bottom-up modelling. We make quantitative predictions, and so go beyond mere correlations (note the correspondence between the x- and y-ranges used in all correlation plots). The test sets are often different in kind from the training sets, as we train only on targets with up to two mismatches in CHAMP and NucleaSeq, but validate on higher mismatched targets. We further validate on *all* mismatched targets in test sets reporting completely different physical quantities (association constants, on rates, off rates, cut fraction, and identified off-targets) for a different sequence library. This makes it impossible to report a straight Pearson correlation between the sets we use for training and validation.

(iii) How does these results fit to previous benchmarks on the same/similar datasets?

Response:

- X. Our scores should not be taken as a benchmarking of models, but only as a test of our model (see also **response V** and the amendment made).

(iv) CFD and uCRISPR are not the most recent models available in the field, and no independent benchmark comparing their performances to those of the latest available models is cited (see comments above).

Response:

- Y. We now refer to Alkan *et al.* as the most recent benchmark, and in response to the reviewers comments have now also included a direct comparison to CRISPRoff.

We have:

- included a comparison to CRISPRoff in **Fig 6**, as well as in **Supplementary Fig S4** and **S5**.

(8) According to the Results section, the method performs well on datasets containing off-targets with more than 2 mismatches. However, in the Supplementary methods section "Parameter estimation" the limit to two mismatches is justified by the fact that "the cleavage activity is much lower for three mismatches and above". (i) To prove that the model is generalizable to off-targets with more than 2 mismatches, and that it does not simply predict them all as negatives, the results/supplement should report the distribution of the predicted cleavage values produced by the method being presented for these off-targets as well as performances evaluated specifically on this subset of the data.

Response:

- Z. We fully agree with the reviewer that we should have included a more thorough evaluation of highly mismatched targets. Since our model puts no limit on the number of mismatches, and indeed does not predict them all as negative, we can make a comparison for up to 20 mismatches. It is true that the highly mismatched the cleavage data (NucleaSeq) is dominated by a single high activity target with more than two mismatches. Our model captures this, resulting in a correlation of 100%. The binding data (CHAMP) has many highly mismatched targets that can be used for validation of the model (correlation 98%, **Figure 2b**), as does the genome wide data.

We have:

- added a new **Figure 2** which details the performance on our test sets. **Figure 2a** show how we quantitatively capture all the highly mismatched targets in the CHAMP library (consisting of block mismatches). **Figure 2b** and **c** show how our model performs on independently gathered datasets from the literature (HiTS-FLIP)

(ii) A range of the other off-target methods use up 6 mismatches as also reported in the literature (Kleinstiver et al. Nat Biotech 2015) and the authors furthermore include analysis of up to 6 mismatches.

Response:

AA. Point well taken, and we now include an analysis of all targets (up to and including 20 mismatches) in **Figure 2** (see also **response Z**).

(9) A section describing the databases used in the study and how the input data looks like is necessary to interpret their usage in the model. The authors should include this.

Response:

BB. We describe what is measured in the CHAMP and NucleaSeq sets (the training data) in the paragraph starting at line 106 but could also have given more detailed information on the sequence library. The whole genome data we use for comparison to off-target classifiers, could also have been better described, and we apologize for this omission.

We have:

- added a description of the shared library in CHAMP and NucleaSeq starting on line 126
- added a **Supplementary Table S2** detailing the whole genome data sets used, together with the guides

(10) A minimum version of the code used in this study should be provided. It was not included in the submission.

Response:

CC. The link to the code (https://gitlab.tudelft.nl/depken_group/SpCas9_kinetic_model_dashboard) was already available through the supplied reporting summary available to the reviewer, but we should also have included the links in a **Code availability** section of the manuscript. The code allows any third party to reproduce the predictions we make, and can be used for benchmarking our model against any other model. We also provide a small dashboard that allows the user to explore cleavage fraction as a function of time in an interactive manner.

We have:

- included a section **Code availability** with direct links to our GitLab page. At the point of publication we also aim to store a snapshot of the GitLab repository permanently at Zenodo.org.

Minor

(11) The scale and unit of measure should be reported in each figure (eg. Fig 1B, 2A, 2B, 3B...).

Response:

DD. **Fig. 1B** is just an illustration, so the scale of the y-axis carries no significance. The x-axis does not need units as it describes discrete states that are all defined in the caption and text.

Fig. 2A, 2B, 3D, 3E, 3H already have the units kBT on the y-axis, and the x-axis is again just a collection of well-defined discrete states, so no unit is assignable.

We have:

- added au (arbitrary units) to the y axis in **Figure 1B**.

(12) The legend and caption of figure 1B are incomplete: are the red dashed lines representing the mismatch at pos. 3 and 15, and not the blue bullets? Also, the abbreviation "sol." is not described (it is later on explained in the caption of Figure 2).

Response:

EE. Agreed, and thanks for pointing this out!

We have:

- included the relevant information in the caption **Figure 1B** and added a legend.

(13) It is a good practice to report the number of data points used to construct box plots, or to scatter the data in addition to the boxes (eg. Fig 3).

Response:

FF. Agreed, and thanks!

We have:

- added a direct reference to the number of points (33) in the text (line 165) and **Figure 3** and **4** captions.
- We now also explicitly show all the data if **Figure 3**, doing away with box plots.

(14) Fig. 4B and C are in the wrong order. The 4 PAM distal mm figure should be 4C, not 4B, according to the caption (and vice versa for 3 PAM distal mm).

Response:

GG. Thanks for spotting this!

We have:

- Changed the order of **Figure 5B** and **C**

(15) The TPR-FPR plots in Supplementary Figure 5 have different scales: this makes them hard to read. Also, a 0-1 scale should be used on both X and Y axis, avoiding unnecessary exponential notation.

Response:

HH. What the reviewer suggests only makes sense on a balanced data set (see also our **response U** above). As our data sets are unbalanced, using linear axis from 0 to 1 would hide any information contained in the plots: all ROC curves would just look like a sharp corner with an AUC very close to 1 (to see this, again note the TPR has risen to unity already around a FPR of 10^{-3} in all cases). This is the reason for us using a *logarithmic scale* for TPR-FPR plots, and opting for precision-recall curves/max F1 scores in the main text.

REVIEWER COMMENTS

Reviewer #2 (Remarks to the Author):

The authors have done a substantial revision and the manuscript has improved, but I'm still confused about their intentions. In the light of the more revised understanding of the manuscript I have a number of concerns.

In the response letter the authors point to line 74 accounting what is new compared to the previous published model. However, I wonder why there is no explicit pointer back to the original work? The authors should make an explicit pointer to the original work stating that this is the new contribution. As I read the purpose both in the manuscript and in response letter, one purpose is to make genome wide activity predictions, but this is not what I'm seeing the software is able to do. Upon testing it the input is a pair, a gRNA and its specified off-target, and then a calculation is provided for this. I do not see a genome-wide scoring of all possible off-targets. The authors need to clarify their intentions and make them consistent with what the software actually provides.

Although the authors include Crisproff they have not included the latest state-of-the-art methods. This includes CRISPR-Net (Lin et al, science advances, 2020) as that paper seems to present the latest advances in off-target prediction. The authors should compare to the latest state-of-the-art methods (including CRISPR-net) before claiming that they have the best performing methods.

What I was hoping for when mentioning CrisprOff was that authors could account for what the kinetics model adds on top of thermodynamic equilibrium. In which cases/situations does it matter and in which cases not? Can this be related to the GC content of the gRNA and off-target etc. The authors should at minimum include a discussion of this.

I'm also confused why the authors want to go up to 20 mismatches. I'm aware that the software can handle it, but what is the biological meaning and do they expect binding to an off-target with 20 mismatches?

Although the authors report higher performance on a precision-recall curve for the HBB gene presented in the main text I do not see any conclusive results in the F1 measure. In the intersection plot the kinetic classifier is best in 1 of 3 cases and for the union none of the methods report impressive predictions and in the case of FANCF where the best performance is obtained by the kinetic classifier, it performs second best. For that case the kinetic classifier is certainly not the best. I do not see what is presented in the main text is representative. In fact the supplementary fig4 contains 8 precision-recall curves (4 intersection and 4 union) and I do not see anything that can justify concluding that the kinetic classifier is the best performing. For example for the union cases including the best one shown in the main text the kinetic classifier is best 2 of 5. In the light of this the authors should

- (i) Present a bar chart of the area under the precision-recall curves.
- (ii) Report objectively which method is superior on which data.

(iii) If there is no clear way to announce “a winner” the authors should modify their statement and just be honest about reaching inclusive results based on the limited test data set.

(iv) state precisely/explicitly what is score here. I assume that the authors “only” score off-targets already pointed out by the experimental data.

For the ROC curve performances reported (sup fig 5) on the individual genes I have a hard time seeing that the kinetic model is superior. For example, as far as I can see the kinetic model is union better on 2 of 4 genes. As authors argue that AUC for all these are close to one, and the reported performances have relatively small differences with relatively big variance. I do unfortunately not see any quantitative evaluation over all test data that justify the claim that the kinetic model is overall better. Taken together the authors should from their results justify how they can call their method for the best one. I do not see that anywhere. Is it for example based on the number of “winning” genes or the accumulated distances to the other methods over the genes, or something else? Please report accurately from the performances obtained.

I am also curious as to why gRNA off-target data from CIRCLE-seq which is in vitro is merged with data from GUIDE-seq which is in vivo into the same unified evaluation for the same gene. As pointed out by (Chung et al, Mol Therapy, 2020) there is a difference in terms of off-target effects and as the authors point out themselves epigenetic modifications are not included in their method. However, since in vitro data is compatible with no modifications and in vivo not I do not see the rationale for fusing them into one grand evaluation for each gene. The authors should elaborate.

In the light of the blurry outcome of the performance evaluation, the relatively few methods compared, and the limitations on GA/AG PAMs, it appear paradoxical that the authors refer to all other off-target scoring in the field as “idiosyncratic”. It would be good style to modify the wordings and be clear on limitations both ways. In the light of this I also wonder what should make researchers use the tool over other tools. Also, from response S do the authors consider the lack of GA/AG PAMs in their off-target assessment to be “compensated” for by other things their method do and therefore a tool researchers should prefer over other tools. I’m obviously concerned that in spite of the lack of taking modifications etc into account that researchers would use the kinetic model over methods that take GA/AG PAMs into account. Can the authors provide arguments for this?

Similarly, if only pre-specified off-targets are given, how do authors envision the software to be applied (even with the other limitations) for off-target assessment prior to selecting suitable gRNAs for an editing experiment?

Upon testing the software, the web interface is fairly fast and responsive, but it is not always clear when a computation is done. It would be helpful to know.

In line with the comments above, the webinterface allows the user to enter guide / (off)-target pairs one-by-one and get the corresponding target cleavage fraction along with several other statistics. From the tool's README it would be possible to construct an program which can run sequential pairs of

guides/targets. However, one would have to be fairly well versed in python to correct the mistakes in the readme and construct a command line tool suitable for large scale evaluation. In addition the tool seems not to offer any way to search for possible off-targets, these would have to come from other sources. With this, how do the authors envision a genome-wide application as stated in the headline (l. 256)?

In their model assumption n.1 (line 119) the authors state that “Mismatch positions are more important than mismatch types (...) and we treat all 12 mismatch types equally.” While according to the current knowledge it is correct to state that mismatch positions are more important than mismatch types, cleavage rates are also known to depend, to a certain extent, on the mismatch identity (Jones et al., “Massively parallel kinetic profiling of natural and engineered CRISPR nucleases”, Nat Biotechnol, 2021). The authors should include some considerations about the impact of this assumption in the discussion. For example according to the data of Jones et al., how much of the variation in the cleavage efficiency data are they not including in the model? Is it negligible?

Since the software does not search for gRNA off-target genome-wide, I’m puzzled why the authors do not apply their software on top of existing off-target predictors to refine the understanding of the critical ones flagged by such software.

We would like to sincerely thank reviewer 2 for taking the time to look at our manuscript once more. We assume that the points no longer raised have been answered to the reviewers satisfaction. For the points that return, we provide additional context and amendments in the hope to satisfy the reviewer.

KEY:

Black: Reviewers original comments

Red: Our responses

Green: Direct quotes from manuscript (underlining only for emphasis here)

Blue: Edits in text

REVIEWER COMMENTS

Reviewer #2 (Remarks to the Author):

The authors have done a substantial revision and the manuscript has improved, but I' still confused about their intentions.

We are very happy the reviewer finds the manuscript improved, but are sorry for the confusion that remains. We hope to clear it up below!

In the light of the more revised understanding of the manuscript I have a number of concerns.

In the response letter the authors point to line 74 accounting what is new compared to the previous published model. However, I wonder why there is no explicit pointer back to the original work? The authors should make an explicit pointer to the original work stating that this is the new contribution.

On the good advice of the reviewer we did include an explicit pointer to our previous proof-of-principle model. To avoid any confusion about the relevant text we recapitulate it below with rearrangements and additions for clarity:

“Base-pairing interactions, protein-DNA interactions⁵³, and induced conformational changes^{51,52,55,56} all contribute to the stability of the Cas9-sgRNA-DNA complex, and we consequently allow for varying gains and losses in the on-target free-energy landscape as the hybrid is extended. These variable gains and losses allow for the formation of metastable states on the on-target, and so consequently constitutes an essential extension of our own previous model for RNA-guided nuclease kinetics⁵¹ with constant gains and losses, as well as of models describing DNA displacement reactions occurring in solution⁵⁷⁻⁶⁰.”

We hope this adequately points to our previous work, and explains its relation to this work.

As I read the purpose both in the manuscript and in response letter, one purpose is to make genome wide activity predictions, but this is not what I'm seeing the software is able to do. Upon testing it the input is a pair, a gRNA and its specified off-target, and then a calculation is provided for this. I do not see a genome-wide scoring of all possible off-targets. The authors needs to clarify their intentions and make them consistent with what the software actually provides.

We choose not to provide a genome wide classification tool because we do not think binary classification is right for this problem. Instead, we suggest using our tool to calculate actual activities, as we show is possible in **Figure 1-2&4**. We only reduce our activity predictor to a classifier to make a point about the importance of including the right physics. We have made one change to the abstract (in blue) to make sure we do not imply we provide a classifier:

“Finally, we show that our quantitative activity predictor can be reduced into a binary off-target classifier that outperforms the established state-of-the-art.”

To make our intent clear the manuscript also includes:

“[...] Our approach overcomes these limitations, and we do not suggest that these benefits should be put by the wayside in order to construct a binary off-target classifier. Still, to strengthen the case for including the full non-equilibrium nature of the problem, we reduce our quantitative kinetic model to a binary classifier (referred to as kinetic classifier) and test how well it performs against three established state-of-the-art off-target predictors.”

The code availability section now reads:

“The code enabling quantitative off-target activity prediction for any guide-target pair is available and maintained on our GitLab page (https://gitlab.tudelft.nl/depken_group/SpCas9_kinetic_model_dashboard). There you will also find a small dashboard application, allowing time resolved activity predictions given a particular sequence and enzyme concentration. A clone of the repository at publication is also available through Zenodo.org.”

Although the authors include Crisproff they have not included the latest state-of-the-art methods. This include CRISPR-Net (Lin et al, science advances, 2020) as that paper seem to present the latest advances in off-target prediction. The authors should compare to the latest state-of-the-art methods (including CRISPR-net) before claiming that they have the best performing methods.

This is a rapidly moving field, and new methods are constantly being added. We happily included CRISPROff when the reviewer previously pointed to this as lacking. With the reviewer raising a new study to consider only at this point, we cannot help but feel the goalpost is being moved once we successfully addressed the reviewer's original concerns. To yet again do the analysis for another off target classifier would lead to unacceptable delays in our minds—especially as we are not providing an off-target classifier.

Strictly, the reviewer is right in that Lin et al Advanced Science (not Science Advances!), 2020 was published (two weeks) before our original submission. Therefore we thought it reasonable to soften our claim of “beating the current state-of-the-art” to “beating the established state-of-the-art”. This still seems apt, considering that Lin et al has attracted little attention so far.

Explicitly we have changed/kept the sentences:

Abstract:

“Finally, we show that our quantitative activity predictor can be reduced to a binary off-target classifier that outperforms the established state-of-the-art.”

Introduction (no change needed):

“[...] we reduce our quantitative activity predictor to a binary off-target classifier that outperforms the leading tools used today^{12,24,29,44}.”

Genome wide off target prediction section:

“[...] and test how well it performs against three established state-of-the-art off-target predictors.”

Discussion:

“The resulting kinetic classifier outperforms established state-of-the-art classification tools on canonical PAM sites in the human genome.”

Again, reducing our model to a genome binary classifier was only done to be able to compare our approach to what models are out there, and not to endorse that approach to the problem. We hoped that by beating the competition we would show the field that incorporating the right (non-equilibrium!) physics will give more predictive power at lower complexity. We think we have succeeded in this, irrespective of what new high complexity models will be introduced in the future.

What I was hoping for when mentioning CrisprOff was that authors could account for what the kinetics model add on top of thermodynamic equilibrium. In which cases/situations does it in particular matter and in which cases not? Can this be related to the GC content of the gRNA and off-target etc. The authors should as minimum include a discussion of this.

We absolutely agree that the set of simplifying assumptions made to our model are only a first step, and we now make a stronger point about this in the discussion:

“Taken together, we have shown that our mechanistic and kinetic model trained on limited bulk data gives biophysical insights into *SpCas9* off-targeting, and that those insights gives quantitative predictive power far beyond the training sets. This predictive power is only expected to improve when including more detailed guide and target sequence features (nucleotide compositions, GC-content or purine/pyrimidine content), as well as alternative PAM sequences into future modelling efforts.”

However, we respectfully disagree with the reviewers implication that it is for us to analyse when the equilibrium assumptions of others are justified. In fact, we question the very approach of picking out specific sequence features (such as e.g. GC content) to determine activity. To quote our manuscript directly:

“Existing physical models^{24,27–29} assume binding equilibration before cleavage, and it remains unclear what predictive power such approaches can ultimately deliver in this non-equilibrium system^{30,31}. To account for the non-equilibrium nature of the targeting reaction, we construct a mechanistic model that captures binding and cleavage reactions in kinetic terms.”

Please also note that there is a supplemental **Section S5** that explains the general implications of the equilibration assumption.

I’m also confused why the authors want to go up to 20 mismatches. I’m aware that the software can handle it, but what is the biological meaning and do they expect binding to an off-target with 20 mismatches?

In our view, the fact that there is little activity on highly mismatched targets should come out of the model, not be stipulated as an extra fact. Consequently we stay away from any preselection, and run our pair-wise (guide-target) activity predictor over all canonical PAM sites.

Although the authors report higher performance on a precision-recall curve for the HBB gene presented in the main text I do not see any conclusive results in the F1 measure. In the intersection plot the kinetic classifier is best in 1 of 3 cases and for the union none of the methods report impressive predictions and in the case of FANCF where the best performance is obtained by the kinetic classifier, it perform second best. For that case the kinetic classifier is certainly not the best. I do not see what is presented in the main text is representative.

First, it is important to be precise about what constitutes a fair comparison between models. Though we do not disagree with the factual statements made by the reviewer, we strongly disagree with the notion that we should compare our model against whichever of the other models happen to perform best on each gene. Which model performs best on a gene can only be determined post hoc, and the results can thus not be called predictions, nor form the basis of any comparison of predictive power.

We are very clear about this in the text:

“More importantly, the kinetic classifier also outperforms the leading off-target predictors for highly-mismatched genomic off-targets of other sgRNAs: performing best on the majority of targets in every pairwise matchup on both union and intersection sets [...]”

More precisely, counting wins by max. F1 scores

Union set:

Kinetic:uCRISPR = 4:1
Kinetic:CRISPRoff = 4:1
Kinetic:CFD = 5:0

Intersection set:

Kinetic:uCRISPR = 2:1
Kinetic:CRISPRoff = 2:1
Kinetic:CFD = 2:1

This is a pretty clear result to us. The fact that our model wins every pairwise matchup is what is stated in the paper, what we believe is relevant, and something we stand by.

In fact the supplementary fig4 contain 8 precision-recall curves (4 intersection and 4 union) and I do not see anything that can justify concluding that the kinetic classifier is best performing. For example for the union cases including the their best one shown in the main text the kinetic classifier is best 2 of 5. In the light of this the authors should

(i) Present a bar blot of the area under the precision-recall curves.

We have now included the AUC of the precision-recall curves in **Figure 6**, and they tell the same story as the max. F1 scores. We have amended the **Figure 6** caption to include the breakdown of wins:

“[...] Matching the models pairwise we can determine which model performs best overall. Using max. F1 scores to count wins on union sets: kinetic:uCRISPR = 4:1; kinetic:CFD = 5:0; kinetic:CRISPRoff=4:1. Using max. F1 scores to count wins on intersection sets: uCRISPR = 2:1; kinetic:CFD = 2:1; kinetic:CRISPRoff=2:1. Using AUC scores to count wins on union sets: uCRISPR = 5:0; kinetic:CFD = 5:0; kinetic:CRISPRoff=3:2. Using AUC to count wins on intersection sets: uCRISPR = 2:1; kinetic:uCFD = 3:0; kinetic:CRISPRoff=2:1. Taken together, the kinetic classifier wins every pairwise matchup irrespective of if we use max F1 or AUC scores, on both union and intersection sets.”

(ii) Report objectively which method is superior on which data.

In our mind we did just this in **Figure 6b**, so it is no clear to us what the reviewer is looking for. Maybe with the added AUC scores it can be seen as more objective? We also performed the analysis based on the Matthews correlation coefficient, and the story remains the same (see below). For ease of presentation we opted to not include it.

(iii) If there is no clear way to announce “a winner” the authors should modify their statement and just be honest about reaching inclusive results based on the limited test data set.

We are very precise with the statement on what constitutes a win (see above), and since the reviewer does not criticize the way in which we define winning, it is hard to defend ourselves against the charge that we are not being honest in what we report.

(iv) state precisely/explicitly what is score here. I assume that the authors “only” score off-targets already pointed out by the experimental data.

We are sorry if this was not clear. We rank every genomic site (human genome GRCh38) with a NGG PAM based on its activity in the low concentration limit, and use experimentally defined off-targets as our true positives. To make it explicit that we compared predictions over the whole genome, we have amended the section:

“To compare our model against the three competitors, we choose to rank all genomic sites based on cleavage activity in the low enzyme-concentration limit. [...] We make our comparison over all canonical PAM sites in the human genome. [...] True positive off-targets are collected from sequencing-based cleavage experiments that used industry-standard sgRNAs and reported multiple off-target cleavage sites^{36–39,41,42} (Supplementary Table S2).”

For the ROC curve performances reported (sup fig 5) on the individual genes I have a hard time seeing that the kinetic model is superior. For example, as far as I can see the kinetic model is union better on 2 of 4 genes. As authors argue that AUC for all these are close to one, and the reported performances have relatively small differences with relatively big variance. I do unfortunately not see any quantitative evaluation over all test data that justify the claim that the kinetic model is overall better. Taken together the authors should from their results justify how they can call their method for the best one. I do not see that anywhere. Is it for example based on the number of “winning” genes or the accumulated distances to the other methods over the genes, or something else? Please report accurately from the performances obtained.

It is indeed based on the number of “winning genes” in pairwise matchups, as explained above and in the text. We hope this is now clear also from the text itself.

I am also curious as to why gRNA off-target data from CIRCLE-seq which is in vitro is merged with data from GUIDE-seq which is in vivo into the same unified evaluation for the same gene. As pointed out by (Chung et al, Mol Therapy, 2020) there is a difference in terms of off-target effects and as the authors point out themselves epigenetic modifications are not included in their method. However, since in vitro data is compatible with no modifications and in vivo not I do not see the rationale for fusing them into one grand evaluation for each gene. The authors should elaborate.

It is also true that the epigenetic context varies among many of the in vivo experiments due to cell type. We take none of this into account, and still outperform the competition. Once we have a model accounting for epigenetics, it would indeed make sense to differentiate the data sets based on this. At present though, we chose to focus on whole-genome analysis including both in vitro and in vivo data sets, and from different cell types, as we do not account for these differences anyway.

In the light of the blurry outcome of the performance evaluation, the relatively few methods compared, and the limitations on GA/AG PAMs, it appears paradoxical that the authors refer to all other off-target scoring in the field as “idiosyncratic”. It would be good style to modify the wordings and be clear on limitations both ways.

We think the outcome is clear (see above). Unfortunately, idiosyncratic was clearly taken as a derogatory term by the reviewer, while we simply meant to say that each model uses a distinct (from each other) measures of activity. We apologise for the unintended confusion. To avoid the audience attaching the same negative valance to the word, we have swapped it out for a near synonym:

“These tools use bioinformatics^{20,21}, machine learning^{22,23}, or heuristic^{12,14,24,25} approaches to rank genomic sites based on distinctive off-target activity scores.”

In the light of this I also wonder what should make researchers use the tool over other tools. Also, from response S do the authors consider the lack of GA/AG PAMs in their off-target assessment to be “compensated” for by other things their method do and therefore a tool researchers should prefer over other tools. I’m obviously concerned that in spite of the lack of taking modifications etc into account that researchers would use the kinetic model over methods that take GA/AG PAMs into account. Can the authors provide arguments for this?

We believe that the binary split into off-targets/not off-targets is deceptive, as there will be a wide range of activities over the genome. For this reason, we do not provide a binary classifier to compete with those on the market. Instead, we offer an quantitative activity predictor.

Similarly, if only pre-specified off-targets are given, how do authors envision the software to be applied (even with the other limitations) for off-target assessment prior to selecting suitable gRNAs for an editing experiment?

We apologise for not managing to make it clear that the whole genome is analysed, and the activity is calculated on each position. To be make sure this is now clear, we modified:

“To compare our model against the three competitors, we choose to rank all genomic sites based on cleavage activity in the low enzyme-concentration limit. [...] True positive off-targets are collected from sequencing-based cleavage experiments that used industry-standard sgRNAs and reported multiple off-target cleavage sites^{S36–39,41,42} (Supplementary Table S2).”

Upon testing the software, the web interface is fairly fast and responsive, but it is not always clear when a computation is done. It would be helpful to know.

A new calculation is performed each time, based on the mismatch content. Precisely what is calculated for the activity prediction is given in the supplemental Equation S2. This equation captures the whole time evolution of the system, allowing us to give both binding and cleavage curves. If desired by the reviewer we could add a little “calculation” complete message to our app.

In line with the comments above, the webinterface allows the user to enter guide / (off)-target pairs

one-by-one and get the corresponding target cleavage fraction along with several other statistics. From the tool's README it would be possible to construct a program which can run sequential pairs of guides/targets. However, one would have to be fairly well versed in python to correct the mistakes in the readme and construct a command line tool suitable for large scale evaluation.

We are sorry if there were mistakes in the readme file. Looking over it again, it is not clear to us what the reviewer is pointing to though. If the reviewer lets us know what is wrong, we would be very happy to fix it!

In addition the tool seem not to offers any way to search for possible off-targets, these would have to come from other sources. With this, how do the authors envision a genome-wide application as stated in the headline (l. 256)?

Again, we are not providing a genome-wide binary classification tool but a quantitative activity predictor. The tools provided can predict the activity at any locus at any time, and applying it sequentially to each genome site you can easily pick out the loci with the highest activity, if you so wish. In the discussion we outline its potential uses for e.g. creating activity titration libraries. We do not provide a classification tool as we do not think splitting targets into binary categories really makes much sense, especially when we have access to the actual activity.

In their model assumption n.1 (line 119) the authors state that "Mismatch positions are more important than mismatch types (...) and we treat all 12 mismatch types equally." While according to the current knowledge it is correct to state that mismatch positions are more important than mismatch types, cleavage rates are also known to depend, to a certain extent, on the mismatch identity (Jones et al., "Massively parallel kinetic profiling of natural and engineered CRISPR nucleases", Nat Biotechnol, 2021). The authors should include some considerations about the impact of this assumption in the discussion. For example according to the data of Jones et al., how much of the variation in the cleavage efficiency data are they not including in the model? Is it negligible?

It is true that some of the present authors have shown that mismatch type does have an effect, though subdominant. To judge if the effect of the mismatch type is negligible, it is not enough to look at the variability in the data, but one also has to establish the effect of this variability in the model predictions. To do this, we would strictly have to include mismatch type in our modelling, and see how much the results are improved. Though interesting, doing this work is clearly outside the scope of the present manuscript.

To clarify that modelling that includes sequence features is definitely a future research direction we include the following in the Discussion:

"This predictive power is only expected to improve when including more detailed guide and target sequence features (nucleotide compositions, GC-content or purine/pyrimidine content), as well as alternative PAM sequences into future modelling efforts."

Since the software do not search for gRNA off-target genome-wide, I'm puzzled why the authors do not apply their software on top of existing off-target predictors to refine the understanding of the critical ones flagged by such software.

Such an analysis might indeed be interesting for people looking for "features" to base top-down classification schemes on, but it is not relevant for the approach we take here. In our minds we do one better than looking for features, and instead explain the activity (high, intermediate, low) in

quantitative terms. This is codified in **Figure 3**, with the rules explained in **Figure 4**. Still, by providing our tool we allow anyone interested to do the type of analysis the reviewer suggest.